# Pyruvate anaplerosis is a targetable vulnerability in persistent leukaemic stem cells

Kevin M. Rattigan[1,5], Zuzana Brabcova[1,5,6], Daniele Sarnello[1], Martha M. Zarou [1], Kiron Roy[1], Ryan Kwan [2], Lucie de Beauchamp[1], Amy Dawson[1], Angela Ianniciello[1], Ahmed Khalaf [1], Eric R. Kalkman [1], Mary T. Scott [1], Karen Dunn [3], David Sumpton [2], Alison M. Michie [3], Mhairi Copland [3], Saverio Tardito [1,2], Eyal Gottlieb [4] & G. Vignir Helgason [1] ✉

Deregulated oxidative metabolism is a hallmark of leukaemia. While tyrosine kinase inhibitors (TKIs) such as imatinib have increased survival of chronic myeloid leukaemia (CML) patients, they fail to eradicate disease-initiating leukemic stem cells (LSCs). Whether TKI-treated CML LSCs remain metabolically deregulated is unknown. Using clinically and physiologically relevant assays, we generate multi-omics datasets that offer unique insight into metabolic adaptation and nutrient fate in patient-derived CML LSCs. We demonstrate that LSCs have increased pyruvate anaplerosis, mediated by increased mitochondrial pyruvate carrier 1/2 (MPC1/2) levels and pyruvate carboxylase (PC) activity, in comparison to normal counterparts. While imatinib reverses BCR::ABL1-mediated LSC metabolic reprogramming, stable isotope-assisted metabolomics reveals that deregulated pyruvate anaplerosis is not affected by imatinib. Encouragingly, genetic ablation of pyruvate anaplerosis sensitises CML cells to imatinib. Finally, we demonstrate that MSDC-0160, a clinical orally-available MPC1/2 inhibitor, inhibits pyruvate anaplerosis and targets imatinib-resistant CML LSCs in robust pre-clinical CML models. Collectively these results highlight pyruvate anaplerosis as a persistent and therapeutically targetable vulnerability in imatinib-treated CML patient-derived samples.

Deregulation of oxidative metabolism has been reported in multiple types of leukaemia[1–3]. Other deregulated metabolic pathways include lipid metabolism, branch chained amino acid metabolism and glutamine metabolism[4–7]. Several studies have examined how metabolic pathways may contribute to resistance to standard therapeutics such as cytarabine or venetoclax, and found that resistance is driven by pathways such as oxidative metabolism, lipid metabolism, nicotinamide metabolism or cellular reduction-oxidation (redox) alterations[5,8–11].

While the tricarboxylic acid (TCA) cycle generates NADH and FADH$_2$ that are used to generate ATP through oxidative phosphorylation within the electron transport chain, intermediates of the cycle are also used for anabolic purposes such as nucleotide synthesis (via aspartate) or fatty acid synthesis (via citrate), necessitating the replenishment of intermediates through input of additional carbons.

Anaplerosis is a metabolic process that refills the TCA cycle with carbons to compensate for those extracted from the cycle

[1]Wolfson Wohl Cancer Research Centre; Institute of Cancer Sciences, University of Glasgow, Glasgow G61 1QH, UK. [2]Cancer Research UK Beatson Institute, Glasgow G61 1BD, UK. [3]Paul O'Gorman Leukaemia Research Centre; Institute of Cancer Sciences, University of Glasgow, Glasgow G12 0ZD, UK. [4]The Ruth and Bruce Rappaport Faculty of Medicine, Technion-Israel Institute of Technology, Haifa, Israel. [5]These authors contributed equally: Kevin M. Rattigan, Zuzana Brabcova. [6]Deceased: Zuzana Brabcova. ✉e-mail: Vignir.Helgason@Glasgow.ac.uk

(cataplerosis) for biosynthetic processes such as nucleotide or fatty acid synthesis. While fatty acid oxidation only supplies 2 carbons to the TCA cycle, glutamine can contribute 4 or 5 carbons (via glutamine oxidation or reductive carboxylation, respectively). During glycolysis, six-carbon glucose is converted to two three-carbon pyruvate molecules. Glucose-derived pyruvate provides 2 carbons to the TCA cycle when pyruvate is metabolised by pyruvate dehydrogenase (PDH; catalysis pyruvate to acetyl CoA reaction), but adds 3 carbons via pyruvate carboxylase (PC; catalysis carboxylation of pyruvate to form oxaloacetate), with both reactions upregulated in primary leukaemia cells[12].

Chronic myeloid leukaemia (CML) is a myeloproliferative disorder that is caused by a reciprocal translocation between chromosomes 9 and 22 t(9;22)(q34;q11) that leads to the formation of the Philadelphia chromosome[13,14]. This translocation generates a constitutively active BCR::ABL1 oncogenic fusion protein[15]. Most CML patients present in chronic phase before inexorably progressing to the more aggressive accelerated phase or lethal blast phase if left untreated[16]. Due to the lack of genetic complexity when compared with other types of leukaemia, and its well-defined leukaemic stem cell (LSC) population (enriched for with CD34+CD38−), chronic phase CML is a paradigm for targeted therapy and thus a model disease to examine how LSCs respond to anti-cancer treatments.

While the use of tyrosine kinase inhibitors (TKIs) such as imatinib (Gleevec) have significantly prolonged patient survival[17], the failure of TKIs to eradicate disease-initiating LSCs means that treatment discontinuation is unsuccessful for the majority of patients[18,19]. In addition, as continual treatment enables more patients to survive longer without progressing to the fatal blast phase, the prevalence of the disease is increasing and CML is expected to become the most common leukaemia type within 30 years[20]. Several explanations for the TKI-resistance of LSCs have been reported. Firstly, mutations either within or outside BCR::ABL1 kinase domain[21], amplification of the BCR::ABL1 gene[22], quiescent cells with higher levels of BCR::ABL1 protein[23], have been reported. Also contributing to TKI-resistance, BCR::ABL1 independent pathways[24–26], suppression of BCR::ABL1 protein level but not gene expression in low oxygen tension typical of the bone marrow niche[27], and contribution of the bone marrow niche cells[26]. Thirdly, enhanced activity of drug exporters has been reported to be critical to TKI-resistance in CML cell lines[28].

Here, using a combination of transcriptomic and stable isotope nutrient tracing dataset analyses, we show that CML LSCs have significantly higher glucose metabolism compared to normal HSCs. Whether these changes persist in the presence of TKI treatment is unknown. Multi-omics analyses of imatinib-treated patient-derived CML CD34+ cells show that TCA cycle activity is significantly altered compared to normal cells. Using uniformly labelled $^{13}$C tracers of the main TCA cycle carbon sources reveals that fatty acid oxidation and pyruvate anaplerosis via MPC1/2 and PC are unaffected by imatinib treatment. We subsequently demonstrate that CRISPR-Cas9 mediated deletion of PC sensitised CML cells to imatinib. Leveraging this metabolic vulnerability, we combine the clinically relevant MPC1/2 inhibitor MSDC-0160[29–31] with imatinib to target patient-derived CML LSCs in vitro and in vivo. Taken together, our work demonstrates the importance of pyruvate anaplerosis for LSC persistence in the presence of TKI, and reveals that MPC1/2 inhibition, with an orally available clinical drug, represents a promising therapeutic approach for CML.

## Results

### Metabolism is highly deregulated in CML LSCs with a bias towards glucose oxidation

To assess whether deregulation in metabolic pathway gene expression occurs in primitive CML cells and translates into differences in metabolite levels requires analysis of transcriptomic[32,33] and liquid chromatography-mass spectrometry (LC-MS) metabolomic datasets[12], comparing patient-derived CML cells to normal human stem/ progenitor cells (Fig. 1a and Supplementary Table 1). The LSC population in CML can be reproducibly characterised using the same cell surface expression markers as HSCs[34]. Interrogation of a transcriptomic dataset (E-MTAB-2581)[33] generated from samples that were enriched for CML LSCs (CD34+CD38−) and normal HSCs (CD34+CD38−), using differential gene expression analysis, revealed an upregulation of genes involved in energy metabolism, including those belonging to fatty acid metabolism (CD36), one carbon metabolism (MTHFD1L) and amino acid uptake (SLC1A5 and SLC7A5) (Fig. 1b and Supplementary Fig. S1A). Gene Set Enrichment Analysis (GSEA) using general Reactome classifiers revealed that metabolism and metabolism-related processes were among the significantly enriched pathways (Fig. 1c). We subsequently examined the ontology of significantly upregulated genes using Panther[35]. Using three different ontology datasets (Molecular function; Biological process; Protein class), significant increases were noted in the fraction of upregulated genes that belong to Catalytic activity, Metabolic process and Metabolite interconversion ontology categories in CML LSC, compared to all genes expressed in either HSCs or LSCs (Supplementary Fig. S1B). We next conducted GSEA on all genes using the Hallmark gene sets. In addition to MYC, p53 and E2F pathways that have previously been identified as dysregulated[33], multiple metabolic pathways display enrichment with the only negatively regulated pathways being KRAS signalling-downregulated genes and Bile acid metabolism (Fig. 1d and Supplementary Fig. S1C). Upregulated pathways include central carbon metabolism pathways: glycolysis, oxidative phosphorylation, and fatty acid metabolism (Fig. 1e). These results agree with previous studies that suggest functional importance of these pathways in myeloid leukaemias[4,5,12,36].

Steady state metabolite analysis from normal and patient-derived samples cultured in physiological Plasmax[37] medium (supplemented as specified in Supplementary Table 2) showed that malate was one of the metabolites with the largest fold change in CML (Supplementary Fig. S1D), in line with the deregulation of the TCA cycle and oxidative phosphorylation observed at the transcriptional level. To accurately assess metabolic activity and specific substrate contribution, we employed uniformly labelled $^{13}$C-tracing analysis of key carbon sources: glucose, palmitate, and glutamine (Fig. 1f). Analysis of $^{13}C_6$ glucose-labelled samples revealed that for all metabolites labelled from glucose, the fraction of labelling was consistently higher in CML CD34+ cells (Fig. 1g and Supplementary Fig. S1E)[12]. While glutamine catabolism was increased in CML CD34+ cells compared to normal CD34+ cells, the difference was not as pronounced as for glucose (lower fold changes; Fig. 1h and Supplementary Fig. S1F). Palmitate contributed fraction of labelling to a slightly higher extent in CML CD34+ cells but this did not reach statistical significance (Fig. 1i and Supplementary Fig. S1G). However, given that the steady state levels of these metabolites are higher in CML cells than normal, increased fatty acid uptake via CD36 and subsequent oxidation may be needed to sustain these higher levels. Finally, we overlaid fraction labelling from glucose and glutamine with transcriptomics data (E-MTAB-2581) for joint-pathway analysis. This underlines that CML has a marked increase in central carbon metabolism both at the transcript and metabolite level (Fig. 1j). Notably, the increases in fraction of labelling was more pronounced from glucose compared to glutamine. Specifically, there was an upregulation of genes involved in glucose uptake (SLC2A1), glycolytic enzyme transcripts and metabolites, as well as TCA cycle enzyme transcripts and metabolites. There was no upregulation in genes encoding enzymes involved in glutaminolysis (GLS and GLUD) but there was an upregulation of glutamine transporters (SLC7A5 and SLC1A5). These findings agree with transcriptomic analyses of murine and human CML studies[38–41]. While CD34+ enrich for progenitors and stem cells, CD34+CD38− enriches further for primitive stem cells. Thus, we

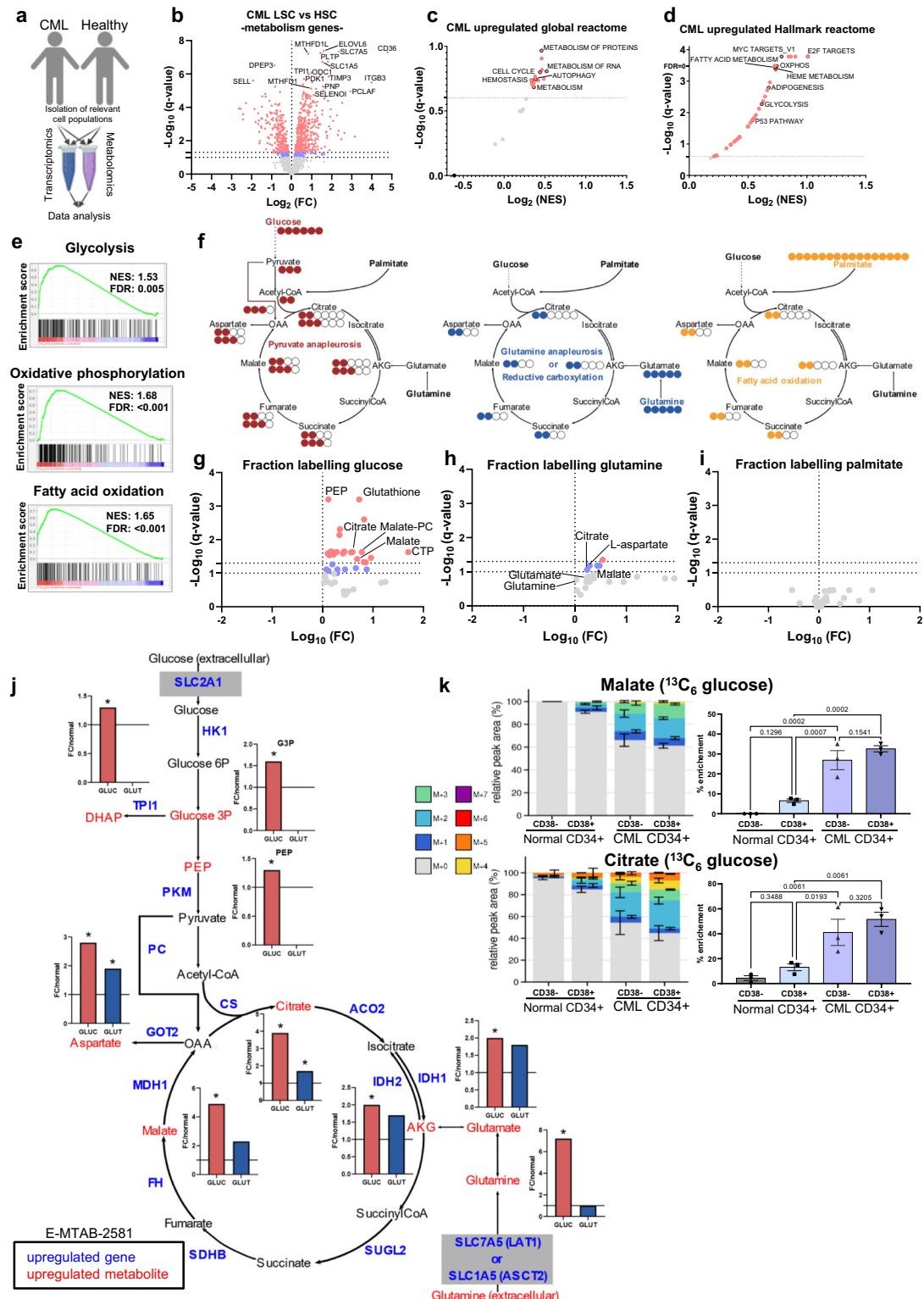

compared the fate of ¹³C-glucose in these two cell populations in normal and CML samples (Fig. 1k). While there was only modest amount of labelling in normal CD34⁺CD38⁻ and CD34⁺CD38⁺ cells, the labelling was significantly increased in both CML primitive populations. Notably there was only a slight, non-significant difference between CML CD34⁺CD38⁻ and CML CD34⁺CD38⁺ cells. Taken together these results demonstrate that CML LSCs are more metabolically active than their normal counterparts and favour glucose to fuel this upregulation.

## Imatinib partially reverses BCR::ABL1-driven reprogramming of CML LSC transcriptome and metabolome

As CML LSCs are more metabolically active than normal HSCs, we hypothesized that at least some of this activity was mediated by BCR::ABL1 kinase activity and would therefore be reversed following TKI treatment. To test this, we generated paired transcriptomics and metabolomic datasets from four CML CD34⁺ patient-derived samples (Fig. 2a). We applied the most commonly used TKI, imatinib, at a clinically relevant dose (2 μM)[42], which trended to inhibiting cell

**Fig. 1 | Human CML LSCs have increased glucose metabolism compared to HSCs. a** Schematic overview of experimental design and omic dataset generation from human CML and non-leukaemic (healthy) samples. Created with BioRender.com (Agreement number: BL25IFA5C3). **b** Volcano plot of differentially expressed metabolism genes comparing CML LSCs and normal HSCs (CD34 + CD38−). Red coloured genes have a *q* value < 0.05 and blue coloured genes have *q* value < 0.1. The Benjamini-Hochberg adjustment was used to correct for multiple comparisons. **c** GSEA conducted using global Reactome pathway gene sets (see source data). **d** GSEA conducted using Hallmark gene sets. **e** CML-upregulated gene sets (from (**d**)) shown are glycolysis, oxidative phosphorylation and fatty acid oxidation. **f** Schematic overview of stable isotope labelling experiments. Tracers employed are $^{13}C_6$ glucose, $^{13}C_5$ glutamine and $^{13}C_{16}$ palmitate. **g–i** Volcano plots of metabolite fraction labelling following 24 h culture in $^{13}C_6$ glucose, (*n* = 4 each group; (**g**)), $^{13}C_5$ glutamine (*n* = 6 CML, *n* = 3 normal; (**h**)) and $^{13}C_{16}$ palmitate (*n* = 3

for each group; (**i**)), comparing CML CD34+ and normal CD34+ cells. Red coloured metabolites have a *q* value < 0.05 and blue coloured metabolites have a *q* value < 0.1. Multiple unpaired *t*-tests were used and the two-stage step-up (Benjamini, Krieger, and Yekutieli) was used to correct for multiple comparisons. (LCMS raw data source: Kuntz et al.[12], for (**g**) and (**i**)). **j** Overlay of fold changes from (**g**) and (**h**) with transcriptomic data from (**b**). Significant changes in gene expression are indicated by gene symbol in blue. Significant changes in fraction labelling are indicated by metabolite name in red. GLUC (red bar) refers to fold change in glucose fraction labelling and GLUT (blue bar) refers to fold change in glutamine fraction labelling with average shown. * Refers to *q* value < 0.05. **k** Fractional labelling (*n* = 3 each normal or CML (and indicated subpopulations)) following 24 h culture in $^{13}C_6$ glucose. Mean and SEM are plotted and an one-way ANOVA was used with multiple comparisons corrected with the Benjamini-Hochberg adjustment. Source data are provided as a Source Data file.

doubling time (*p*-value = 0.0659; Supplementary Fig. S2A) in agreement with previous studies[43]. It is possible that this increase is due to imatinib inducing a modest, sample-dependent increase in the level of apoptosis (*p*-value = 0.0809; Supplementary Fig. S2B). Differential gene expression revealed upregulation of 452 genes (2.8%) and downregulation of 576 genes (3.6%), including several genes that encode proteins that regulate metabolic pathways (Supplementary Fig. S2C, S2D). A significant increase in the negative regulator of cell cycle, p21 (*CDKN1A*) was observed, and downregulated genes included those involved in cell cycle progression *CDK1* and *TPX2*, indicative of cell cycle arrest. At a protein level, albeit in different samples, the response in CDK1 levels varied in a sample dependent manner, while phospho-CRKL, which is immediately downstream of BCR::ABL1 signalling, was decreased by imatinib in the two patient samples tested (Supplementary Fig. 2O). Interestingly there was no change in CD36 (Supplementary Fig. S2D) which is highly upregulated in CML LSCs compared to normal HSCs (Supplementary Fig. S1A). Furthermore, no changes were detected in glutaminase or any of the glutamine transporters. These results were cross-referenced with a patient dataset (E-MTAB-2594), obtained from CML CD34+ cells treated with imatinib for 7 days[44], where a higher proportion of expressed genes were significantly different (10,762: 56%) (Supplementary Fig. S2E). GSEA on generated transcriptomic data demonstrated that, while immune response pathways, adipogenesis and p53 pathway were upregulated in response to imatinib (Supplementary Fig. S2F, S2H), the most downregulated pathways included oxidative phosphorylation, glycolysis, and fatty acid oxidation (Fig. 2b, d). Notably, similar results were obtained following longer (7 days) imatinib treatment (Fig. 2c, e, and Supplementary Fig. S2G, S2I), including consistent downregulation of oxidative phosphorylation in both datasets (Fig. 2d, e). Collectively, these data show that TKI-mediated BCR::ABL1 inhibition causes downregulation of key metabolic pathways, although not sufficiently to prevent LSC survival, suggesting that additional metabolism-targeting drugs may be required to effectively eradicate CML LSCs. However, the effect of imatinib on metabolism should be considered when adding metabolic inhibitors to imatinib treatment, with the preference to target deregulated pathways unaffected by imatinib or aiming to completely shut down pathways that are only partially inhibited by imatinib.

To obtain a global overview of metabolism during TKI treatment, we performed steady state analysis on imatinib-treated primary CML CD34+ cells, focusing on metabolites that can be reproducibly detected in primary samples (Fig. 2f). A significant increase in AIR with a corresponding non-significant decrease in AICAR likely reflects a decrease in purine synthesis related to decreased proliferation (Supplementary Fig. S2J). While we detected a modest sample dependent increase in AMP/ATP ratio there was no significant change in ATP levels (Fig. 2g and Supplementary Fig. S2K). Mitochondrial respiratory capacity (measured by FCCP-induced oxygen consumption rate) was reduced in imatinib-treated samples (Supplementary Fig. S2L). At a

protein level, AMP-activated protein kinase (AMPK)[45] and phospho-AMPK levels were variable depending on patient sample (not detected in CML#9; Supplementary Fig. S2O). These variances may also be due to differences in time-point used for western blots compared to RNAseq (24 h vs 48 h). Notably, the more potent TKI, nilotinib has been previously shown to increase phospho-AMPK in CD34+ CML cells[46]. We also detected decreases in oxidative stress and cellular redox (Fig. 2h, i and Supplementary Fig. S2M) and TCA cycle metabolites (Fig. 2j and Supplementary Fig. S2N). To examine pathways in an unsupervised manner, joint pathway analysis was conducted on data from paired RNA-seq and LC-MS samples. The most significantly affected pathways were pyruvate metabolism and the TCA cycle (Fig. 2k, l). Overall, the results of the integrated omic analysis are consistent with TKI treatment causing a reduction of the main CML central carbon metabolism pathways.

## Imatinib disrupts nutrient contribution to TCA cycle and redox metabolites in primitive CML cells without affecting MPC1/2 levels or PC activity

Given the reduction in central carbon metabolism observed following TKI treatment, we explored how CML cells maintain lower levels of metabolism while supporting survival. Using patient-derived CML CD34+ cells we examined the effect of imatinib on TCA cycle substrates: glucose, palmitate, and glutamine as well as on their contribution to the TCA cycle intermediates. To obtain a global overview of the relative substrate contribution to TCA cycle, and how this is altered by imatinib, the percentage of carbon pool from each tracer was merged to give the nutrient contribution plots of TCA cycle metabolites (Fig. 3a and Supplemental Fig. S3A). Analysis using a two-way ANOVA revealed that palmitate and glutamine contribute a similar fraction of labelling to the TCA cycle metabolites (*p* > 0.1) except to citrate (*p* < 0.0001) and glutamate (*p* = 0.02), with palmitate contributing more to the former and glutamine more to the latter. For all TCA cycle-related metabolites, there was less contribution from glucose (*p* < 0.05). However, this may be affected by the high palmitate concentration (100 μM) required for tracing experiments, resulting in an underestimation of glucose contribution. Indeed, the addition of 100 μM palmitate had a significant effect on the contribution of glucose, which can be seen in the comparison to the much higher contribution from glucose seen in medium where the only source of fatty acids is from lipid-rich Albumax II (compare Supplementary Fig. S3B and S3C).

Examining each dataset in detail, we found that glucose and glutamine contributions were reduced following imatinib treatment, while palmitate contribution was not significantly different (Fig. 3b–d). Decreases in glucose contribution were observed in TCA cycle metabolites (citrate, malate and aspartate), serine and glycine biosynthetic pathways (serine and glycine) and Cahill cycle (L-alanine). Decreases in labelling of TCA cycle metabolites and a decrease in the labelling of redox metabolites, glutathione, and glutathione disulfide (GSSG), was

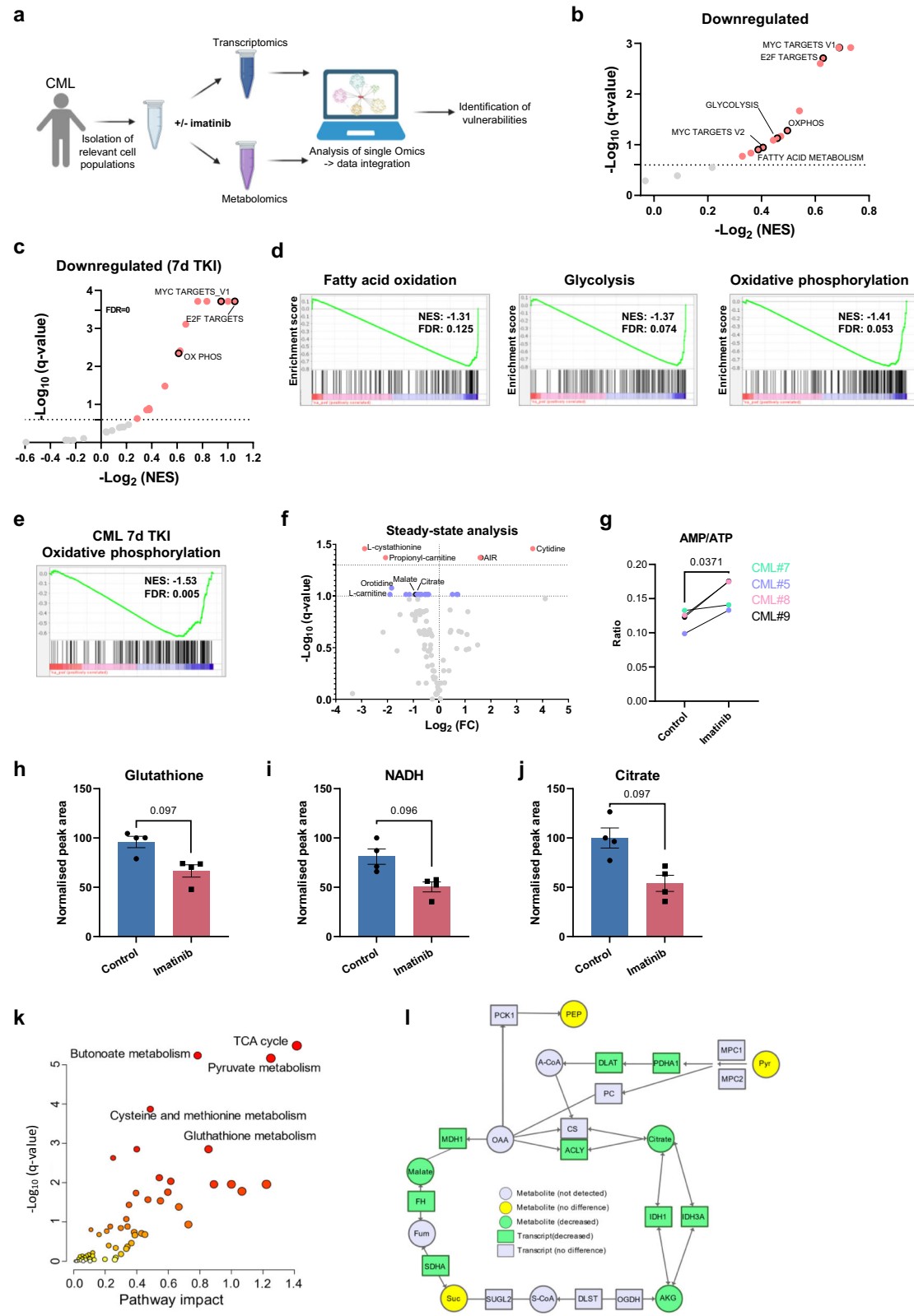

also observed in $^{13}C_5$ glutamine samples. Further analysis showed that, for most metabolites, the decrease in glucose occurred to a greater extent than the decrease in glutamine fractional contribution (Fig. 3e). Notably, PC relative to PDH contribution of glucose was increased in imatinib-treated cells, measured by the ratio of m + 3 ($^{13}C_3$) to m + 2 ($^{13}C_2$) labelling in malate, citrate, and aspartate (Fig. 3f). This is due to a more pronounced and consistent decreases in the PDH-derived

fraction (m + 2; Supplementary Fig. S3D) without a significant decrease in the PC-derived (m + 3) fraction (only 2 out of 4 cases showed a consistent decrease in m + 3), hence the increase in relative activity (PC/PDH) in all cases. These results are in agreement with paired transcriptomic data showing downregulation of *PDH* but no change in *MPC1/2* or *PC* levels following imatinib treatment (Fig. 2l). Similar results were seen following nilotinib treatment, with reduction in total

**Fig. 2 | Imatinib partially reverses BCR-ABL driven metabolic reprogramming of human CML cells. a** Schematic overview of paired omic dataset generation from human CML cells (n = 4) exposed to imatinib. CD34+ cells were treated in vitro with 2 μM imatinib for 48 h. Created with BioRender.com (Agreement number: UV25I-FAAWP). **b** GSEA conducted on data described in (**a**) using Hallmark gene sets. The downregulated gene sets are shown. **c** GSEA conducted using Hallmark gene sets on CML CD34+ cells treated with imatinib for 7 days. The downregulated gene sets are shown. **d** Imatinib-downregulated gene sets (from (**b**)), shown are fatty acid oxidation, glycolysis and oxidative phosphorylation. **e** Imatinib-downregulated gene sets (from (**c**)), shown is oxidative phosphorylation. **f** Volcano plot comparing metabolite steady state levels between control and imatinib treated CML CD34+ cells (n = 4 patient-derived samples, 2 independent experiments). Cells were treated with 2 μM imatinib for 48 h. Red coloured metabolites have q value < 0.05 and

blue coloured metabolites have a q value < 0.1. Multivariate analysis was carried out using MetaboAnalyst 5.0. The Benjamini-Hochberg FDR method was used to correct for multiple comparisons. **g** AMP/ATP ratios are shown (n = 4 patient-derived samples). Statistical analysis performed using a paired t-test. **h–j** Selected metabolites from (**a**) with corresponding q values shown. Plots were generated using Autoplotter. Average and SEM are plotted. **k** Joint pathway analysis of transcriptomics and metabolomics data from control and imatinib treated CML CD34+ cells (n = 4 patient-derived samples, 2 μM imatinib for 48 h). Analysis was carried out using MetaboAnalyst 5.0, and hypergeometric test, degree centrality and combined queries were selected. The Benjamini-Hochberg FDR method was used to correct for multiple comparisons. **l** The TCA cycle from F is shown and significant changes denoted in the key. Source data are provided as a Source Data file.

metabolite levels and $^{13}C$-glucose incorporation into TCA cycle metabolites, suggesting that the effect is mediated through inhibition of BCR::ABL1, but not other kinases affected by imatinib such a as c-Kit (Supplementary Fig. S3F–G).

Changes in intracellular metabolism can be driven by changes in metabolite uptake and secretion. Thus, we quantified extracellular flux of spent medium samples from imatinib-treated and untreated primary CML CD34$^+$ cells using a YSI bioanalyser. Analysis of these data showed that imatinib treatment decreased glutamine consumption (Supplementary Fig. S3G). While imatinib-treated samples consumed less glucose than control cells, when corrected for a decrease in proliferation this difference disappeared. It is noteworthy that the cell density used here was designed to maintain metabolic steady state and avoid depleting nutrients, so it is possible that differences in glucose consumption were below the assay detection limit.

Overall, these data suggest that PC-mediated pyruvate anaplerosis persists in imatinib-treated CML CD34$^+$ cells and may contribute to residual TCA cycle activity. To investigate the relevance of these findings to the therapy resistant CD34$^+$CD38$^-$ population we quantified the effect of imatinib on $^{13}C$-glucose labelling in sorted cells (Fig. 3g). While imatinib caused a statistically significant reduction in glucose incorporation in CML CD34$^+$CD38$^-$ cells, the relative PC/PDH activity increased in all three patient samples in this primitive population with a trend towards statistical significance (Fig. 3h, p-value = 0.0870).

Given that orally bioavailable TKIs provide life-long disease control for the majority of CML patients, it is important to consider the toxicity of available inhibitors for fatty acid metabolism, the absence of anaplerosis from fatty acids, as well as the high levels of fatty acid oxidation observed in normal LSCs when prioritising targetable metabolic pathways. We therefore aimed to target pyruvate anaplerosis in CML LSCs.

## PC activity can be abrogated using an MPC1/2 inhibitor in primitive CML cells

As it is possible that $^{13}C_3$-labelled TCA metabolites could be the product of cytoplasmic maleic enzyme 1 (ME1) activity[47], we decided to use inhibitors of MPC1/2 to prevent pyruvate entry into mitochondria, allowing for distinction between cytosolic and mitochondrial metabolites (Fig. 4a). While the most commonly used research-grade MPC1/2 inhibitor is the tool compound UK-5099, MSDC-0160 is of relevance as it has successfully undergone Phase 2 clinical trial for diabetes (NCT00760578). We screened several concentrations of these compounds in the K562 CML cell line. This is critical as the albumin present in either FBS or in Albumax II, which is present in medium for primary sample culture, has been shown to bind each drug, reducing their bioavailability and necessitating higher concentrations for complete inhibition in cell culture compared to Seahorse assay that does not comprise FBS[48]. While both compounds led to a concentration-dependent reduction in glucose contribution to fraction of labelling as previously reported[48], we observed the persistence of the m + 3 fraction in aspartate and malate which indicates the presence of

cytosolic ME1 activity, while there are reductions in all other fractions (UK5099: Supplementary Fig. S4A, MSDC-0160: Supplementary Fig. S4B). Furthermore, these results indicate that alpha-ketoglutarate (AKG) and AKG-derived glutamate are a more appropriate approximation of mitochondrial glucose oxidation for this cell line, with both inhibitors reducing levels of all labelled fractions in these metabolites in a concentration-dependent manner.

We next tested MSDC-0160 treatment on primary CML samples, which notably express higher transcript levels of both *MPC1/2* and *PC* (Fig. 4b, c). Steady state analysis in CML CD34$^+$ cells only showed significant changes at a metabolic level at the highest dose of 100 μM (Fig. 4d). Unlike that noted in K562 cells, MSDC-0160 reduced both total glucose contribution and m + 3 contribution to TCA cycle metabolites in a dose-dependent manner in primary CML CD34$^+$ samples (Fig. 4e and Supplementary Fig. S4C). Using 20 μM (achievable dose in patients) or 100 μM MSDC-0160 caused significant decreases to both m + 2 and m + 3 labelled fractions unlike imatinib which only affected m + 2 fractions (Fig. 4e compared to Supplementary Fig. S3D), with variable effects on TCA cycle metabolite m + 3/m + 2 ratios (Supplementary Fig. S4D). While the dose required to fully inhibit labelling is relatively high (100 μM), this is consistent with high FBS binding of the drug as previously reported[48]. This inhibition was also achieved in the presence of stromal cells (Supplementary Fig. S4E). Moreover, we confirmed that inhibition of glucose oxidation was achieved in the primitive CD34$^+$CD38$^-$ population, to a similar extent as in the CD34$^+$CD38$^+$ progenitor population (Fig. 4G). These results demonstrate that upregulated PC-mediated pyruvate anaplerosis in CML LSCs, which is less affected by imatinib than PDH-mediated pyruvate oxidation (Fig. 2l), can be targeted with MSDC-0160. Interestingly *PC* expression strongly associates with survival in adult AML with high-risk cytogenetics (Fig. 4f) with the difference just dropping outside significance threshold when excluding the M3 FAB subgroup (Supplementary Fig. S4F).

Taken together these data demonstrate that glucose oxidative metabolism is distinct in primary CML CD34$^+$CD38$^-$ cells compared with normal CD34$^+$CD38$^-$ cells, and mitochondrial pyruvate metabolism can be inhibited with a clinically relevant MPC1/2 inhibitor.

## PC ablation or inhibition of mitochondrial pyruvate import sensitises CML cells to imatinib in vitro

To test the functional impact of PC inhibition, we generated a PC knockout K562 CML blast crisis cell line using CRISPR-Cas9 technology (Fig. 5a). *PC* deletion did not affect proliferation of K562 cells (Fig. 5b). LC-MS was then applied to investigate if *PC* deletion reduced glucose contribution to TCA cycle metabolites. Using AKG and AKG-derived glutamate as readouts of mitochondrial pools, we observed a marked reduction in the m + 5 fraction (Fig. 5c). This m + 5 fraction is derived from the condensation of PC-m + 3 and PDH-m + 2[49]. The m + 2 fraction was unchanged or slightly increased while m + 3 fractions were slightly decreased (Supplementary Fig. S5A–B), which is likely due to persistent cytoplasmic maleic enzyme activity seen in Supplementary

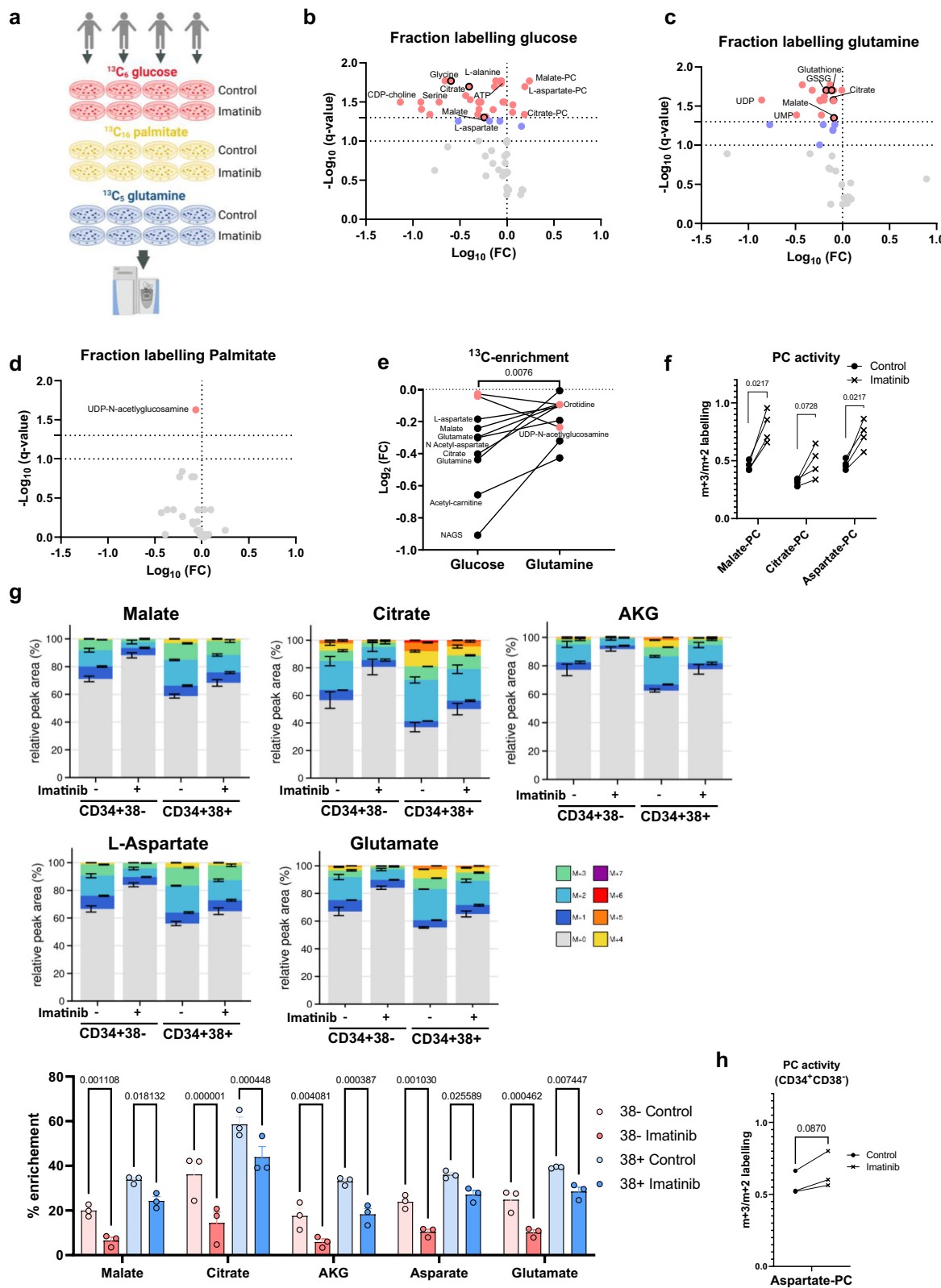

Fig. S4A–B. To test the effect of TKI on *PC* deletion, control and knockout cell lines were treated with imatinib and cell viability measured. While PC ablation sensitised K562 to imatinib in both viability and colony-forming cell (CFC) assays, this was not observed using the FDA-approved protein translational inhibitor omacetaxine mepesuccinate (used on occasions for advanced phases of CML), suggesting selective increased reliance on pyruvate anaplerosis in TKI treated cells

(Fig. 5d, e). Supporting this, knockout of PC in the KCL22 cell line, which have lower levels of PC, failed to increase sensitivity to imatinib (Supplementary Fig. S5C–F). As there is no clinically applicable PC inhibitor available and MPC1/2 inhibition should replicate its effect, we next tested the impact of combining MSDC-0160 with imatinib on colony formation potential of CML CD34+ cells isolated from six CML patients. While imatinib reduced the number of colonies, there was a

**Fig. 3 | Imatinib disrupts nutrient contribution to TCA cycle in CML CD34+ cells. a** Schematic overview of experimental design. Created with BioRender.com (Agreement number: RE25IF9ZLB). **b–d** Volcano plots of metabolite fraction labelling from $^{13}C_6$ glucose (**b**), $^{13}C_5$ glutamine (**c**) and $^{13}C_{16}$ palmitate (**d**) comparing control and imatinib treated CML CD34+ cells ($n = 4$ patients-derived cells per group, 2 µM imatinib for 48 h). Red coloured metabolites have a *q* value < 0.05 and blue coloured metabolites have a *q* value < 0.1. Multiple paired *t*-tests were used and the two-stage step-up (Benjamini, Krieger, and Yekutieli) was used to correct for multiple comparisons. **e** Fold changes (imatinib/control) for fraction labelling from $^{13}C_6$ glucose and $^{13}C_5$ glutamine in CML CD34+ cells ($n = 4$ patients-derived cells per group, 2 µM imatinib for 48 h). In red are the two metabolites with larger decreases in $^{13}C_5$ glutamine contribution. Statistical analysis was performed using a

single paired *t*-test (two-sided). **f** For relative PC/PDH activity CML LSCs (CD34+, $n = 4$ patients-derived cells per group, 2 µM imatinib for 48 h), the m + 3/m + 2 ratio is shown for the indicated metabolites. Multiple paired *t*-tests (two-sided) were used and the two-stage step-up (Benjamini, Krieger, and Yekutieli) was used to correct for multiple comparisons. **g** Fractional labelling from $^{13}C_6$ glucose for indicated cell fractions ($n = 3$ patients-derived cells for each group, 2 µM imatinib for 24 h). Average and SEM are plotted, and a two-way ANOVA was used for statistical analysis, *p*-values were not corrected for multiple comparisons. **h** Relative PC activity calculated in aspartate in CML LSCs, 2 µM imatinib for 24 h. Statistical analysis was performed using a single paired *t*-test (two-sided). Source data are provided as a Source Data file.

further concentration-dependent reduction by the addition of MSDC-0160 (Fig. 5f). We then tested the effect of this combination on primary normal CD34$^+$ cells to ensure that it is not toxic due to off-target effects. Encouragingly, no significant effect of single agent or combination treatment was observed on the CFC potential of these cells (Fig. 5g). The above data therefore support the rationale of combining a mitochondrial pyruvate transport inhibitor with imatinib in vivo.

### Inhibition of MPC1/2 targets human CML LSCs in vivo
To assess the clinical relevance of these findings we employed a patient-derived xenograft model, transplanting CML CD34$^+$ cells into sub-lethally irradiated immunocompromised NRG-W41 mice[50] (Fig. 6a). While the low level of engraftment typical of chronic phase CML CD34$^+$ cells does not allow for survival experiments to be conducted, this model is the gold-standard to assess human LSC survival due to the long-term engraftment and well-defined stem cell population. Eight weeks post-transplant, mice were randomly assigned to groups for 4 weeks of treatment: vehicle, imatinib (100 mg/kg/day), MSDC-0160 (30 mg/kg/day) and combination of imatinib and MSDC-0160. All treatment arms were well tolerated, indicated by consistent body weight throughout the study (Fig. 6b). At the experiment end point, bone marrow cells were collected, and engraftment was determined by flow cytometry (Supplementary Fig. S6A). No changes were observed in engraftment of human CD45$^+$ leukocytes (Supplementary Fig. S6B) and in the percentage of CD34$^+$ cells when comparing vehicle or imatinib arms to the combination (Supplementary Fig. S6C). However, when examining the most primitive and clinically relevant CML population (CD34$^+$CD38$^-$), there was a marked reduction in both the percentage and absolute number of human CML LSCs following treatment with MSDC-1060 (Fig. 6c, Supplementary Fig. S6D). Critically this reduction was maintained in the presence of quiescence-inducing imatinib, which, as previously shown, fails to target CML LSCs in vivo[12,33]. Thus, the combination of MPC1/2 inhibition with standard-of-care drug imatinib causes a significant reduction in therapy resistant CD34$^+$CD38$^-$ CML cells at clinically administrable doses in vivo.

### Discussion
Transformation from a normal cell to a cancerous one requires metabolic changes to fuel bioenergetic demands required for increased proliferation or adaptation to a hostile environment[51]. Metabolic adaptation may also occur during anti-cancer treatment. For example, AML cells have been shown to exhibit transient metabolic adaptation in vivo, driving resistance to chemotherapy[5,52]. An important consideration is that rapid proliferating normal cells also undergo metabolic reprogramming. For example, pyruvate metabolism has previously been shown to inhibit clonal expansion of normal clonogenic haematopoietic progenitor cells, which necessitates the inclusion of normal cells in metabolic studies[53].

In CML, the constitutively active BCR::ABL1 kinase drives leukaemic proliferation through stimulation of several key cell survival pathways frequently activated in cancer. Recently, upregulation in key metabolic pathways such as central carbon metabolism has also been

demonstrated in CML cells, including therapy-resistant CML LSCs, which is required to facilitate and sustain CML LSC survival[12,41]. However, the role or existence of these cytoprotective changes following acute or prolonged exposure to TKI treatment in clinically relevant cell population is not well understood and requires detailed metabolic analysis of primary human samples. Transcriptomic analyses revealed a significant deregulation of multiple metabolic pathways in chronic phase CML LSCs when compared with normal HSCs, suggesting BCR::ABL1-mediated metabolic reprogramming in the early phase of leukemogenesis. This was also apparent following analysis of stable isotope tracing of TCA cycle substrates. Specifically, when compared to normal cells, primitive CML cells have significantly higher glucose metabolism, slightly higher glutamine metabolism and only modest differences in fatty acid metabolism. Interestingly, IL-3 has been shown to promote glucose transport and metabolism while BCR::ABL1 (or RAS) can mimic this signalling and is IL-3 dependent[28,54–58]. Overlay of transcriptomic data[32,33] with metabolomic data (generated here and previously[12]) confirmed that CML LSCs have increased central carbon metabolism at both transcript and metabolic levels. Specifically, there was an upregulation of glucose transporter (*SLC2A1*) and glycolytic enzyme transcripts. Interestingly, there was no upregulation of transcripts of glutaminolysis enzymes (*GLS* and *GLUD*) but there was an upregulation of glutamine transporters (*SLC7A5* and *SLC1A5*). Glutamine transporter upregulation driven by c-MYC has been reported previously[59–61], but whether deregulated c-MYC[33] drives this upregulation in CML LSCs warrants further investigation.

While metabolic reprogramming offers multiple targets for therapeutic intervention, we focused on pathways that remain deregulated in the presence of imatinib, as its clinical use in CML is so ubiquitous it can be considered as a baseline. To model the effect of TKI treatment, we isolated and cultured patient-derived CML CD34$^+$ cells in physiological culture medium, in the absence or presence of imatinib-mediated BCR::ABL1 inhibition and subjected them to RNA-seq and LC-MS mediated metabolomics to generate paired multi-omics datasets. We subsequently used isotope-assisted metabolomics and uniformly labelled $^{13}C$ tracers of the key TCA cycle carbon sources glucose ($^{13}C_6$), glutamine ($^{13}C_5$) and palmitate ($^{13}C_{16}$) and showed that the fractional contribution of pyruvate anaplerosis via PC and fatty acid oxidation are the only metabolic pathways not decreased (reverted) following imatinib treatment. While deregulated PC activity has been shown to be important in other malignancies[62,63], its importance to leukaemia is unknown.

As mitochondrial and cytosolic metabolite pools can confound the interpretation of isotope tracing experiments, we utilised MPC1/2 inhibitors (UK-5099 and MSDC-0160) to confirm that pyruvate carboxylation occurs in the mitochondria. In addition, we demonstrated that CRISPR-Cas9 mediated deletion of PC sensitises CML cells to imatinib. Leveraging this metabolic vulnerability, we combined MSDC-0160 with imatinib to target CML LSCs in vitro and in vivo (Supplementary Fig. S7). In conclusion, our data highlights a strong bias towards glucose metabolism in CML LSCs and upregulation clinically relevant downstream target, mitochondrial pyruvate metabolism.

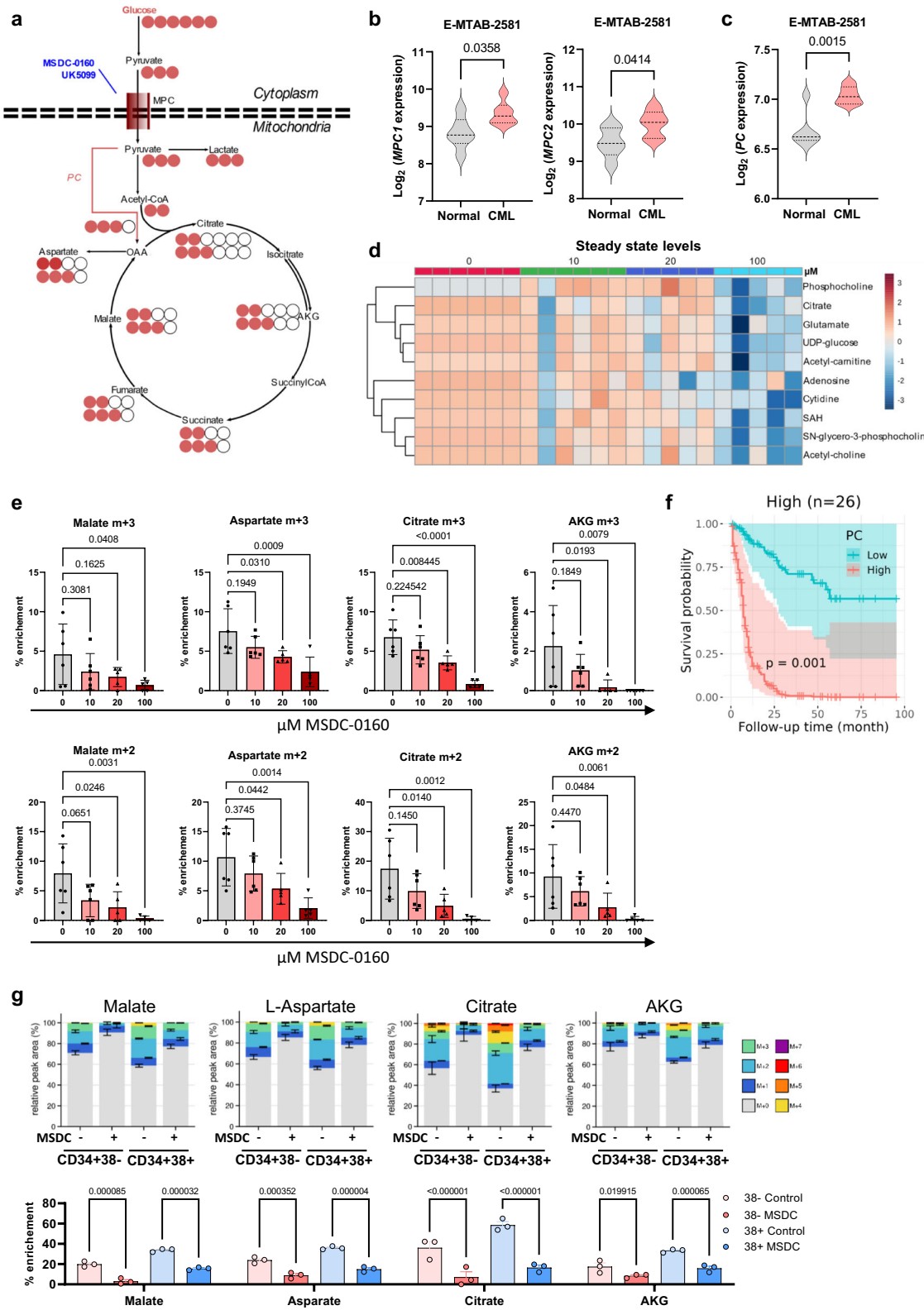

In terms of potential limitations and future work, while our experimental design aims to examine nutrient contribution in a physiological setting and shows that palmitate (100 μM) is a major contributor to TCA cycle, it is important to state that the non-assigned fraction is unknown. In addition, a wide range of human plasma free palmitate concentrations has been reported (HMDB: 25–1000 μM), even within the same study using different methods

$(66 \pm 9.9$–$122 \pm 48 \, \mu M)$[64]. The variability of reported palmitate levels, the potential effect from non-labelled palmitate from plastic, and the absence of other types of lipids means that Supplementary Fig. S3A may not present an accurate reflection of relative nutrient importance. In addition, genetic ablation of *PC* and or *MPC1/2* in patient-derived LSCs would be informative but this is impractical due to rapid differentiation of patient-derived samples in vitro.

**Fig. 4 | CML CD34+ cells have increased PC activity that can be abrogated using MPC1/2 inhibitor. a** Schematic of $^{13}C_6$ glucose conversion into $^{13}C_3$ pyruvate and its subsequent transport into mitochondria via MPC1/2 where it can be metabolised by pyruvate carboxylase or pyruvate dehydrogenase. Inhibition of MPC1/2 by UK-5099 or MSDC-0160 are indicated. **b, c** Violin plots showing expression of (**b**) *mitochondrial pyruvate carrier 1 and 2 (MPC1* and *2)*, (**c** *pyruvate carboxylase* (*PC*) in normal HSCs and CML LSCs (CD34 + 38−) in indicated dataset. For **b, c**, an unpaired *t*-test (two-sided) was used for statistical analysis. **d** Steady state levels of significantly different metabolites (*p* < 0.1) in CML CD34+ cells (patients-derived cell samples; *n* = 6 for 0 μM and 10 μM, *n* = 5 μM for 20 μM and 50 μM) exposed to indicated concentrations of MSDC-0160 for 24 h. Analysis was carried out using MetaboAnalyst 5.0. A one-way ANOVA was performed, and the Benjamini-Hochberg FDR method was used to correct for multiple comparisons. **e** Individual patient-derived samples from (**d**) are plotted for statistical analysis. Average and SEM are plotted (patients-derived cell samples; *n* = 6 for 0 μM and 10 μM, *n* = 5 μM for 20 μM and 50 μM). An ordinary one-way ANOVA was used with Dunnett's test used to correct for multiple comparisons. **f** Survival plot showing effect of *PC* expression on overall survival in patients with high-risk cytogenetics, stratified by low or high (20th and 80th percentiles) *PC* expression levels. The 95% confidence intervals were represented by the boundaries of mean ± 1.96 * standard deviation, n referrers to number of patients. The effect of the interaction term between *PC* and high cytogenetic risk was estimated by Cox proportional hazards model (HR = 3.423, CI = [1.652, 7.094], *z* = 3.310, two-sided *p* = 0.000934). **g** Fractional labelling from $^{13}C_6$ glucose for indicated cell fractions (*n* = 3 patients-derived cells for each group, 50 μM MSDC-0160 for 24 h). Average and SEM are plotted, and a two-way ANOVA was used for statistical analysis, *p*-values were not corrected for multiple comparisons. Source data are provided as a Source Data file.

Furthermore, the precise mechanism by which *MPC1/2* and *PC* is upregulated in CML LSCs and persists in the presence of TKI treatment is unclear. Finally, whether this TKI-escape mechanism exists in other malignancies warrants additional investigations.

In terms of clinical relevance, CML represents a paradigm for targeted therapy in cancer with the potential for cure. While the introduction of TKIs have revolutionised treatment of CML, a small but important minority of patients fail to respond[65]. CML LSCs are inherently insensitive to TKI treatment, and it is within this cellular fraction that drug resistance and disease progression evolve. As CML is managed by orally available TKI treatments in the majority of patients, testing medication that requires parenteral administration is not practical. MSDC-0160 has been shown to specifically inhibit MPC1/2 activity in a variety of cell types[31]. It belongs to a class of thiazolidinediones (also called glitazones), and has already completed a promising Phase II trial for type 2 diabetes (NCT00760578; NCT01103414)[29] and for patients with Alzheimer's disease (NCT01374438)[30]. Here we have shown that the clinically relevant and orally available MPC1/2 inhibitor, MSDC-0160, could be both efficacious and practical to combine with TKI to improve treatment responses in CML patients.

## Methods

### Ethical approval

For primary CML patient-derived samples, cells were obtained from peripheral blood or leukapheresis product. All samples were from patients that were in chronic-phase at the time of diagnosis. All patients gave written informed consent in agreement with the Declaration of Helsinki and with the approval of the National Health Service (NHS) Greater Glasgow and Clyde Institutional Review Board. Ethical approval was granted to the research tissue bank (REC 15/WS/0077) and for using surplus human tissue in research (REC 10/S0704/60). These results are not sex specific. While more male patient samples were used (17) than female (8) this was due to the amount of cells available for each patient in biobank. The median age at diagnosis of patients was 47-years old, ranging from 24 to 70 years old.

Animal experiments were performed in accordance with Home Office regulations and under approved project licence PPL PP2518370 and personal licence PIL IE2DD924E.

### Primary samples

Primary CML samples were sourced from CML patients either from 50 mL peripheral blood or leukapheresis product. Patients were in chronic phase CML at the time of diagnosis and gave informed consent in accordance with the Declaration of Helsinki and approval of the National Health Service (NHS) Greater Glasgow and CLyde Institutional Review Board. The CD34$^+$ cells were isolated using the CliniMACS (Miltenyi Biotec) and purity verified to be >95% by flow cytometry (*Apoptosis and CD34 analysis*). Non-leukemic cells were isolated from femoral head material using human CD34 MicroBeads (Miltenyi Biotec), according to the manufacturer's instructions. Purity was verified by flow cytometry to be >90%.

### Statistical analyses

No statistical methods were used to determine sample size. For experiments, a minimum of four samples was used to give adequate power. The investigators were not blinded to sample or treatment during experiments. Non-parametric data was analysed using Kruskal–Wallis test.

### Patient information

Please see Supplementary Table 1 for available patient information.

### Reagents

Imatinib mesylate was purchased from LC Laboratories (I-5508). A stock solution of 10 mM was prepared in sterile Milli-Q water and stored at 4 °C. MSDC-0160 (Apex Biotechnology: B3702) UK5099 (Merck: PZ0160-5MG) and Omacetaxine mepesuccinate (ChemGenex Pharmaceuticals) were made up in DMSO. Stock concentrations were 50 mM for MSDC-0160, and 10 mM for UK5099 and Omacetaxine.

### Cell culture

Primary CML samples were thawed and recovered overnight in Plasmax, a physiological cell culture medium[37]. This medium was supplemented with nutrients and growth factors as described in Supplementary Table 2, then filter sterilised through a 0.2 μM filter (Fisher Scientific: 10509821). For cell line experiments, the medium was RPMI with 10% dialysed FBS and 1% penicillin/streptomycin. For tracing experiments 11 mM $^{13}C_6$ glucose was added to glucose-free RPMI. Cells were counted and viability calculated using a CASY counter with the following parameters to gate live cells: E-cur 7.5-22.5, N-cur 5.25-20.5. Doubling time over 48 h was calculated by dividing the cell density into four times the seeding density ($4 \times 4 \times 10^5$cells/mL = 16). For stromal co-culture experiment, irradiated M2-10B4 and S1/S1 mouse cell lines that are genetically engineered to express human cytokines were seeded ($8 \times 10^4$ cells each in 1000 μL) in DMEM supplemented with 10% FBS and hydrocortisone onto collagen-coated plates. The following day, medium was removed and $2 \times 10^5$ primary CML resuspended in 1000 μl of Plasmax (13-UC-glucose), seeded on top of the feeder cell layer for 24-h culture in absence or presence of 50 μM MSDC-1060. Non-adherent cells were then harvested, and samples prepared for LC-MS. All cell lines were obtained form DSMZ and regularly tested to ensure they were free of contamination such as mycoplasma.

### RNA extraction

RNA was extracted from 200,000 CML CD34$^+$ cells using an RNA easy mini kit (Qiagen) according to the manufacturer's instructions.

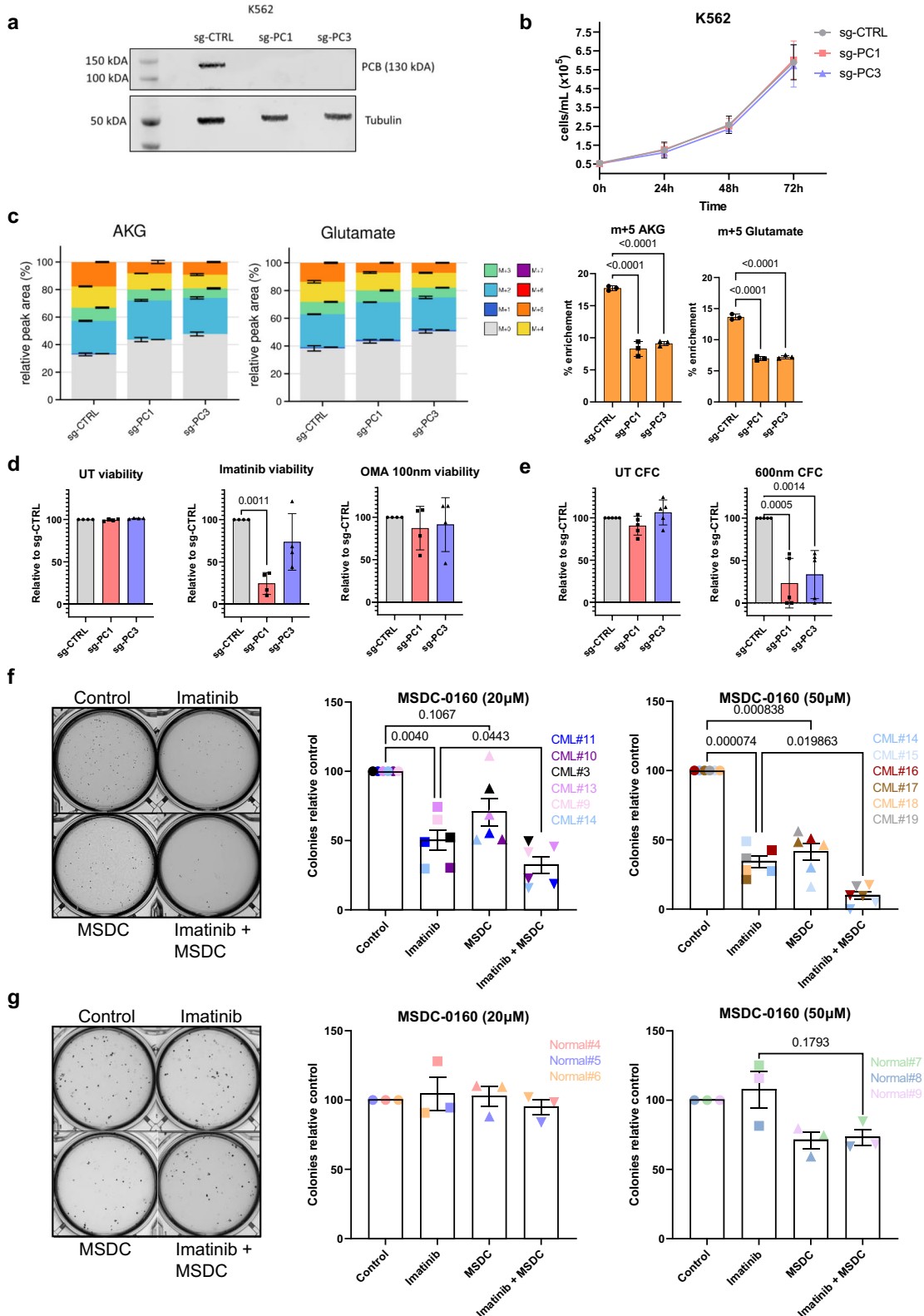

## RNA-seq

Libraries were generated with the TruSeq Stranded mRNA LT Kit (Illumina) and ran on the Illumina Next-Seq 500 using the Illumina High-Output 75 cycles kit (2 × 36 cycles, paired end reads, single index). FastQ files were prepared with bcl2fastq (v. 2.20.0.422, Illumina). QC, alignment, and parsing of files into count matrices was performed in command line, with subsequent differential gene expression (DGE) analysis performed in R (1.2.1335). Adapter trimming was performed using Scythe (version 0.981), and Sickle (version 0.940), was used to trim bases with quality scores of <20. Prior to and after this pre-processing fastqc (version 0.11.2) was run to ascertain sequence quality, alongside the efficacy of the pre-processing steps. Trimmed reads were indexed and aligned using Hisat2 (version 2.1.0). Hisat2 indexes (GRCh38 genome_tran) were

**Fig. 5 | PC ablation or inhibition of mitochondrial pyruvate import sensitises CML to imatinib. a** Western blot of PC and loading control (tubulin) from vector control or PC knockout K562 cells. **b** Proliferation of K562 cells from (**a**). Shown is average and SD from 3 independent experiments. **c** Fractional labelling from $^{13}C_6$ glucose into indicated metabolites of PC knockout cells are shown ($n = 3$ independent cultures) after 24 h. Plots were generated using Autoplotter, average and SD are plotted. Statistical analyses were conducted on the m + 5 fractions. An ordinary one-way ANOVA was used with Dunnett's test used to correct for multiple comparisons. **d** Cells viability was measured using Annexin V and 7AAD. Cells were either left untreated (UT) or exposed for 72 h to 600 nM imatinib or 100 nM omacetaxine (OMA). Shown is average and SD from $n = 4$ independent experiments. An ordinary one-way ANOVA was used with Dunnett's test used to correct

for multiple comparisons. **e** Colony forming cell (CFC) potential of control and PK knockout cells. Cells were either left untreated (UT) or exposed for 72 h to 600 nM imatinib. Shown is average and SEM from $n = 5$ independent experiments (one replicate each time). An ordinary one-way ANOVA was used with Dunnett's test used to correct for multiple comparisons. **f, g** Representative images of colonies and bar plots of colony numbers resulting from CML CD34+ cells ($n = 6$ CML patient-derived samples; (**f**)) or normal CD34+ cells ($n = 3$ healthy donors; (**g**)) exposed to 2 μM imatinib, 20 μM or 50 μM MSSC-0160 or combination for 72 h. Average and SEM are plotted. A repeated measure one-way ANOVA was used with multiple comparisons corrected for using Tukey's test. Source data are provided as a Source Data file.

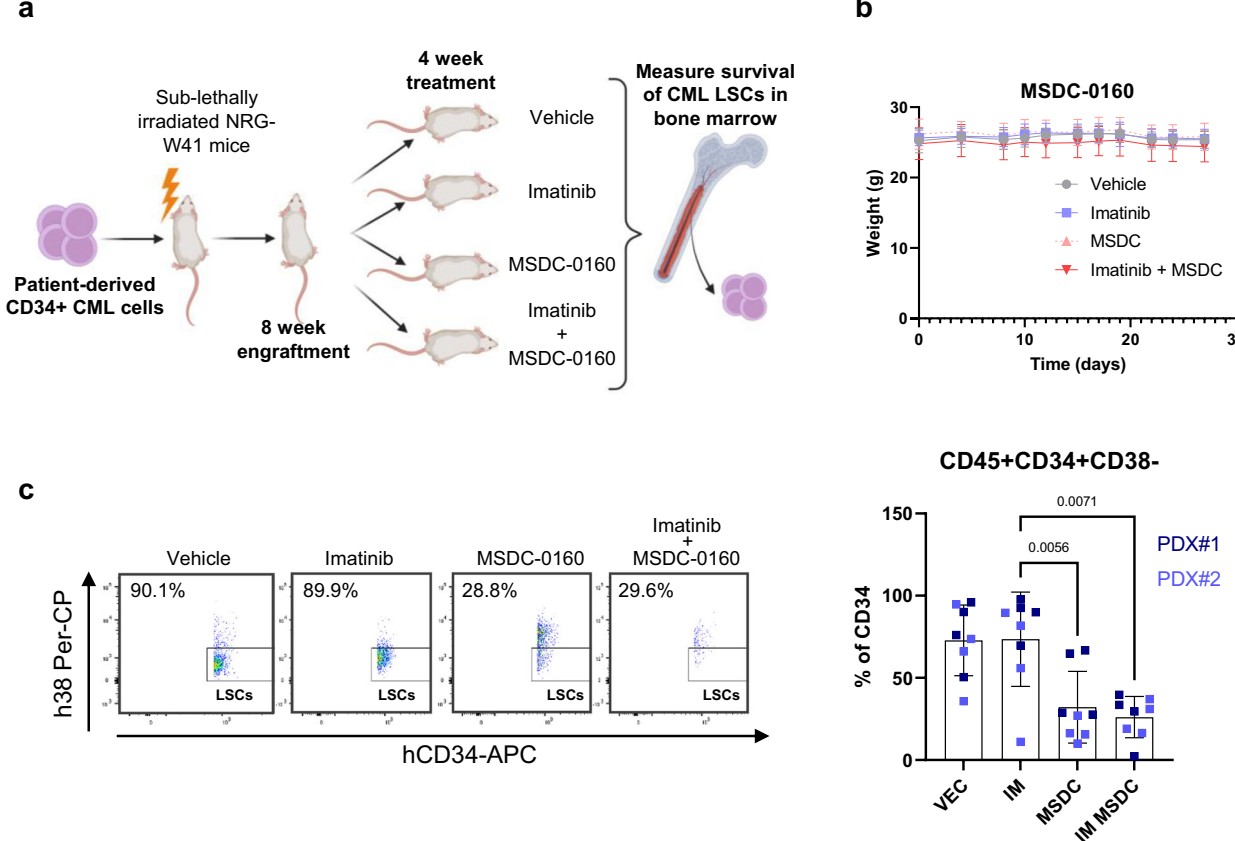

**Fig. 6 | Combining inhibition of Mitochondrial pyruvate transport and BCR-ABL eradicates CML stem cells in vivo. a** Schematic overview of experimental design for in vivo experiment. Created with BioRender.com (Agreement number: WM25IF8MNY). **b** Weights of mice taken over experiment ($n = 4$ mice per group;

2 independent PDX experiments). The average and SD are plotted. **c** The percentage CML CD34 + CD38− LSCs is shown ($n = 4$ mice per group; 2 independent PDX experiments). Average and SD are plotted. A Kruskal–Wallis test was used to analyse data. Source data are provided as a Source Data file.

obtained from the John Hopkins Center for Computational Biology, 2020. Samtools (version 0.1.19044428 cd) view was used to convert the resulting.sam to.bam files, whilst samtools sort was used to sort the.bam files. Assembly was achieved through the use of stringtie (John Hopkins Center for Computational Biology, 2020), with output.gtf files converted to count matrices using the python script prepDE.py (stringtie version 1.3.3b.Linux_x86_64). Reads were assembled using an annotated reference human genome (GRCh38.p13), obtained from GENCODE (GENECODE, 2020). DESeq2 (version 1.26.0) was used to generate results sets from the gene and transcript count matrices. G genes with read counts too low to allow for the calculation of p and adjusted *p*-values (padj: Benjamini-Hochberg) were removed from the data sets leaving gene and transcript counts of sizes 16,069 and 45,218 respectively. Microarray analysis was carried out in R studio (version 1.1.4).

### Microarray analysis
Data were analysed using Limma (version 3.34.9).

### GSEA analysis
GSEA (version 4.1) was conducted on pre-ranked lists (ranked by pi score calculated by multiplying LOG fold change by -LOG (corrected *p*-value)).

### Palmitate conjugation for stable isotope tracing experiments
For conjugation of palmitate, a 20 mM stock was made in 100% EtOH. This was incubated on a shaking heater, 60 °C until the solution clarified. The palmitate was then added to 10% BSA (ultra-fatty acid free in EBSS, Roche, 03117057001) to give a palmitate:BSA ratio of 1:3 BSA. This solution was left for 15 min in water bath (37 °C), and then added to final concentration in complete medium.

## LC-MS sample preparation

Cells were counted, and viability calculated using a CASY counter with the following parameters: E-cur 7.5-22.5, N-cur 5.25-20.5. Timepoints were chosen to ensure equal viability within patient-derived samples for between experimental arms. Cell number was multiplied by peak volume (size) and this number was used to calculate volume of solvent to extract in. Cells were pelleted by centrifugation ($400 \times g$, 10 min, room temperature) at which medium was removed for YSI analysis. Cells were then washed twice with ice cold PBS (Calcium and Magnesium free: Thermo Fisher Scientific) with pulse centrifugation ($12,000 \times g$, 15 s, 4 °C) used between and after washes. Cells were then extracted in LC-grade ACN:MeOH:H$_2$O solvent (−20 °C, 5:3:2) by disrupting the pellet by pipetting followed by a 5 second vortex. Finally, debris was pelleted by centrifugation ($16,000 \times g$, 10 min, 4 °C), supernatant transferred to LC-MS glass vials that were stored at −80 °C until analysis.

## LC-MS

The LC system composed of a ZIC-pHILIC column (SeQuant, 150 × 2.1 mm, 5 μm, Merck KGaA) with a ZIC-pHILIC guard column (SeQuant, 20 × 2.1 mm) with an UltiMate 3000 HPLC system (Thermo Fisher Scientific). The aqueous mobile-phase solvent was 20 mM ammonium carbonate-0.1% ammonium hydroxide solution and with acetonitrile being used for organic mobile phase. A linear biphasic LC gradient was conducted from 80% organic to 80% aqueous for 15 min for a total run time of 22 min. The column temperature was maintained at 45 °C flow rate set to 200 μL/min. The MS used in this study was a qExactive Plus Orbitrap Mass Spectrometer (Thermo Fisher Scientific) operating in polarity switching mode. The MS set up was calibrated using a custom CALMIX in both ionization modes before analysis and a tune file targeted towards the lower $m/z$ range was used. Full scan (MS1) data was acquired in both ionization modes in profile mode at 70,000 resolution (at $m/z$ range 75–1000), an automatic gain control (AGC) target of $1 \times 10^6$ (max fill time of 250 ms), with spray voltages +4.5 kV (capillary +50 V, tube: +70 kV, skimmer: +20 V) and −3.5 kV (capillary −50 V, tube: −70 kV, skimmer: −20 V) and s-lens RF level of 50 for the front optics. The capillary temperature 375 °C, probe temperature 50 °C, sheath gas flow rate 25, auxiliary gas flow rate 15a.u. 15 and sweep gas flow rate of 1a.u. The mass accuracy achieved for all metabolites was below 5ppm. Data acquisition was achieved with Thermo Xcalibur 4.3.73.11 software.

## LC-MS analysis

The peak areas of different metabolites were determined using Tracefinder 4.1 software (Thermo Fisher Scientific). Metabolites were identified by accurate mass of the singly charged ion and by known retention times on the pHILIC column. A commercially available standard compound mix (Merck: MSMLS-1EA) had been analysed previously on our LC-MS system to determine accurate ion masses and retention times. The 13 C labelling was determined by quantifying peak areas for the accurate mass of all isotopologues of each metabolite.

## Steady state analysis

Two independent experiments were performed, data normalised to first sample in each experiment, processed through Autoplotter (version 2.3D)[66], samples as experiments, and normalised data as replicates. These data were processed on Metabonalyst using the Rlog transformation and mean-centring to ensure data followed a normal distribution. For MSDC-0160 treated samples a batch correction (to non-drug control) was used prior to analysis using Metabonalyst. Volcano plots were generated, and an FDR adjusted $t$-test threshold 0.1 was utilised. Note both fold changes and $p$-values are log transformed. Top-25 scoring metabolites are included in heatmap.

## Analysis of stable isotope tracing experiments

Biological and technical replicates were processed through Autoplotter (version 2.3D)[66] and natural abundance was corrected for. Fraction of enrichment was calculated as m + 2 and above. For CML samples treated with or without imatinib, multiple paired $T$-tests were conducted on Log10 transformed values in Graphpad Prism. The correction for multiple comparisons was the two-stage step-up method of Benjamini, Krieger and Yekutieli with an FDR (Q) value of 10%. For volcano plots Log10 FC vs Log10 ($q$ value) are plotted with $q$ values > 5% in red and >10% in blue. For normal vs CML comparisons, the process was the same with the exception that unpaired t-tests were used. For PC activity in normal vs CML, this is denoted by fraction m + 3. For imatinib vs untreated we look at relative activity to PDH (m + 3/m + 2 ratio) and conducted multiple $t$-tests with correction for multiple comparisons being the two-stage step-up method of Benjamini, Krieger and Yekutieli with an FDR (Q) value of 10%. To examine contribution to carbon pool we corrected contribution of each isotopologue by the number of carbons it contributes to a given metabolite, i.e., for glutamine contribution from m + 5 is greater than m + 3. Note that for cell line experiments standard deviation is shown.

## Bioanalyser

A YSI 2900 biochemistry analyser was used as per the manufacturer's instructions to quantify glucose, lactate, glutamine, and glutamate in the media. The concentration of each metabolite was normalised to cell number and the rate of uptake or secretion per hour was calculated relative to cell free medium. The exchange rate per 48 h [secretion (+) or consumption (−)] for a specific metabolite (x) was obtained according to the following equation:

$$\frac{\Delta metabolite}{(cell\ number\ day\ 0 + cell\ number\ day\ 2)/2}$$

where Δmetabolite = ((x) mmol spent medium − (x) mmol cell-free medium).

## Oxygen consumption rate measurements (Seahorse analysis)

Seahorse (XF96) assay was conducted as previously described[12]. CMLCD34+ were treated for 24 h with 1 μM imatinib prior to analysis.

**Lentivirus production.** Lentiviruses for pLentiCRISPRv.2 were produced by the calcium phosphate method using pCMV-VSV-G (envelope plasmid: RRID: Addgene_8454) and psPAX2 (packaging constructs: RRID: Addgene_12260) vectors and human embryonic kidney (HEK) 293FT cells for transfection.

## Multi-omic analysis

The joint pathway tool of Metabonalyst[67] was used for analysis of paired data. Here we used the hypergeometric test for enrichment analysis, the topology measure was degree centrality, and the integration method was combined queries.

## Gene ontology analysis

Panther[35], (version 17.0) was used for mapping to different sets and Fisher's exact test used to compare fraction of number of upregulated genes per pathway/total number of upregulated genes with same fraction of all expressed genes.

## Survival analysis

A multivariable Cox proportional hazards model was fitted to TCGA data in R software. Two datasets were analysed, one including and the other excluding the FAB M3 subtype. To simplify the model, a backward stepwise model selection procedure was applied to the complex Cox survival model, which originally included age, sex, FLT3_ITD, protocol, transplant_type, PC, FAB, and cytogenetic_risk predictors

(full model). The reduced model (reduced model) was obtained by retaining age, FLT3_ITD, protocol, transplant_type, PC, and cytogenetic_risk predictors, while dropping the interaction term between PC and FAB, from the original model.

The resulting models can be represented as:

Full model: *proportional hazard ~ age + sex + FLT3_ITD + protocol + transplant_type + PC * (FAB + cytogenetic_risk)*

Reduced model: *proportional hazard ~ age + FLT3_ITD + protocol + transplant_type + PC * cytogenetic_risk*.

## Western blot analysis

Cells were lysed in RIPA buffer containing mini-Complete protease inhibitor cocktail and phosphatase inhibitors (both Roche). Total protein concentration was quantified using a Pierce BCA kit (Thermo Fisher Scientific: 23227). Equal amounts of protein (5–7.5 μg) were heated at 95 °C for 5 min and separated (120 V) in 4–12% gels (Novex) for SDS-PAGE. Proteins were transferred onto PDVF membranes (Thermo Fisher Scientific: 21882) then blocked in 2% BSA (in Tris-buffered saline, 0.01% Tween (TBS-T)) for one hour. Next, membranes were incubated overnight at 4 °C with the primary antibodies. The primary antibodies used were p-AMPK (Cell signalling, Cat #2531, diluted 1:1000), AMPK (Cell signalling, Cat #2532, diluted 1:1000), p-CRKL (Cell signalling, Cat #3181, diluted 1:500), CDK (Cell signalling, clone POH1, Cat #9116, diluted 1:1000), PC (Proteintech Cat# 16588-1-AP) and H3 (Active Motif, clone MABI 0301, Cat #39763 diluted 1:1400). The membranes were rinsed three times with TBS-T, then incubated with secondary HRP-linked antibodies; Anti-rabbit IgG HRP-linked Ab (Cell Signalling Cat#7074, diluted 1:10,000) and Anti-mouse IgG HRP-linked Ab (Cell Signalling Cat#7076, diluted 1:10,000), for 1 hour at room temperature. The SuperSignal West Femto Maxi detection system was used (Thermo Fisher Scientific: 34095) and imaging was carried out using a LI-COR Odyssey Fc gel-doc system. All uncropped and unprocessed scans are in Source Data File.

## Apoptosis and CD34 analysis

Cells were stained with Annexin V (fluorescein isothiocyanate (FITC, BioLegend: Cat# 640906, 5 μL/test), 7-AAD (BD Bioscience: Cat# 559925, 5 μL/test) and CD34+ (APC, BD Bioscience, clone 581, Cat# 555824, 2 μL/test) in 50 μL Hanks' Balanced Salt Solution (HBSS) for 20 min (room temperature in dark). CML cells were analysed by flow cytometry (BD FACSVerse, BD FACSuite version 1.0.6.5320) and data were analysed using FloJo (version 10).

## CRISPR-Cas9 mediated deletion

To target the human *PC* gene, guides were designed using the optimized tool https://www.genscript.com/gRNA-database.html. Two guides were chosen and ordered from Integrated DNA Technologies. These were annealed and cloned in Bsmb I–digested lentiCRISPRv.2-puro (RRID: Addgene_52961). After stable integration of lentiCRISPRv.2 using lentiviral transfection and 1-week selection using puromycin (2.5 μg/ml), guides were validated by performing Western blotting. Oligonucleotides from IDT are as follows:

g1 forward: CACCGCAGGCCGCGGCCGATGAGAT
g1 reverse: AAACATCTCATCGGCCGCGGCCTGC
g2 forward: CTGAAGTTCCGAACAGTCCA
g2 reverse: TGGACTGTTCGGAACTTCAG
g3 forward: CACCGACAGGTGTTCCCGTTGTCCC
g3 reverse: AAACGGGACAACGGGAACACCTGTC

## Patient derived xenografts experiments

For in vivo engraftment, 1 million CML CD34+ cells, were transplanted via tail vein into sublethally irradiated (2.5 Gy) female NRG-W41 (Jackson Laboratory: RRID:IMSR_JAX:026014, NOD.Cg-*Rag1^{tm1Mom}Kit^{W-41J}Il2rg^{tm1Wjl}*) mice aged 8-10 weeks. Female mice are used as this gives higher engraftment. For housing, light is 12 h

light and 12 h darkness, temperature was 20–24 °C, humidity was 45–65% and feeding (LBS biotech: SDS Cat# SDS801730) and water were ad libitum. Eight weeks following transplant, drug treatment was started with both imatinib (100 mg/kg/day (50 mg/kg BID); oral gavage) and MSDC-0160 (30 mg/kg; oral gavage once daily) given for 4 weeks. At the end of treatment, bone marrow cells were collected. This was done by placing inverted cut leg bones into 0.5 mL Eppendorf tubes with holes at bottom. These in turn were placed within 1.5 mL Eppendorf tubes containing PBS, centrifuged (12,000 × g, 20 s). Cells were stained (300 μl/test) with anti-mouse (APC-Cy7 BD Biosciences, clone 30-F11, Cat# 557659, RRID: AB_396774, 1 μl), anti-human CD45 (FITC; BD Biosciences, clone HI30, Cat# 555482, RRID: AB_395874, 10 μl), anti-human CD34 (APC; BD Biosciences, clone 581, Cat# 555824, RRID: AB_398614, 2 μl) and anti-human CD38 (PerCP; BioLegend, clone HIT2, Cat# 303520, RRID: AB_893313, 2 μl) antibodies for 20 min in dark (room temperature) prior to flow cytometry analysis as described above. This model does not result in high disease level, hence for animal welfare, post-irradiation sickness or tolerability of treatment were the main factors monitored for.

## Fluorescence-activated cell sorting (FACS)

For isolation of the CD34+CD38− cell population from normal or CML samples, CD34+ cells were stained with anti-human CD34 and anti-human CD38 as described above and sorted using a FACSAria Fusion Cell sorter (BD Biosciences, BD FACSDiva Software v8.0.1).

## Reporting summary

Further information on research design is available in the Nature Portfolio Reporting Summary linked to this article.

# Data availability

The expression profiling RNA-seq data generated in this study have been deposited in public Gene Expression Omnibus (GEO) database under accession code GSE216837. The publicly available datasets used in this study are available in the EMBL-EBI database under accession code E-MTAB-2581. LCMS data for analysis of patient samples is in source files. To protect patient privacy, raw LCMS files generated in this study are available upon request to the corresponding author immediately once approval of biobanks ethical approval panel is granted and access will not be time limited. The LCMS samples will be maintained long-term (>10 years) and raw LCMS files will be maintained indefinitely (>10 years on institutes network drive, Redundant Array of Independent Disks (RAID)). Additional information concerning human samples can be obtained from the corresponding author. Source data are provided with this paper. The remaining data are available within the Article, Supplementary Information or Source Data file. Source data are provided with this paper.

# Code availability

Relevant code has been deposited on GitHub and made publicly available https://github.com/Kiron-J-Roy/Pyruvate-Anaplerosis-is-a-Targetable-Vulnerability-in-Persistent-Leukaemic-Stem-Cells?.

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

## Acknowledgements

We would like to thank the Core Services and Advanced Technologies at the Cancer Research UK Beatson Institute (C596/A17196; A31287). We thank Connie J Eaves for providing NRG-W41 mice, Jerry Colca for advice regarding in vivo administration of MSCD-0160 and Simone Cardaci for discussion regarding interpretation of the results. We thank all patients and healthy donors who donated samples and the National Health Service (NHS) Greater Glasgow and Clyde Biorepository; Alan Hair for sample processing; Tom Gilbey for cell sorting and the CRUK Beatson Institute mouse facility staff for housing of mice and treatment-xenograft experiments. Schematic graphs in Figs. 1a, 2a, 3a, 6a, S7A were generated using BioRender (https://biorender.com). We would like to acknowledge and pay special tribute to our colleague Zuzana Brabcova for her significant contribution. This work was funded by The Kay Kendall Leukemia Fund (KKL1069), Blood Cancer UK (formerly Bloodwise; Ref 18006), The Howat Foundation, Cancer Research UK (C57352/A29754), Tenovus Scotland, Cancer Research UK Glasgow Centre (A25142), Friends of Paul O'Gorman Leukaemia Research Centre (all to G.V.H.); NHSGGC Endowment Fellowship (GN17ON425) to K.M.R.; Cancer Research UK Glasgow (A23982) to S.T. and Cancer Research UK Glasgow core funding (A31287) to D.S.[2], S.T. and the Cancer Research UK Beatson Institute.

## Author contributions

K.M.R., Z.B., and G.V.H. developed the concept and designed the experiments. K.M.R. and Z.B. analysed data and performed all experiments. D.S1., K.R., M.T.S., M.M.Z., L.B., A.D., A.I., A.K., and K.D. assisted with ex vivo and in vivo studies, including data analysis. Z.B., D.S1. and K.R. assisted with statistical analysis. A.M.M. and M.T.S. assisted with in vivo studies. D.S2., E.R.K., A.M.M., E.G., S.T. and M.C. provided technical or material support. K.M.R. and G.V.H. wrote the manuscript and all other authors reviewed it. G.V.H. supervised the work.

## Competing interests

M.C. has received research funding from Cyclacel and Incyte, is/has been an advisory board member for Novartis, Incyte, Jazz Pharmaceuticals, Pfizer and Servier, and has received honoraria from Astellas, Novartis, Incyte, Pfizer and Jazz Pharmaceuticals. All other authors declare that they have no competing interests.
