## [Peer Review File · Nature Communications]

Reviewers' Comments:

Reviewer #1:

Remarks to the Author:

In the manuscript by Rattigan et al., the authors have followed up on the role of metabolism and anaplerosis for the progression of BCR-ABL1-driven CML. The authors also show that an inhibitor of mitochondrial pyruvate carboxylase plus imatinib reduces CML LSC.

The manuscript is well written and addresses a long-overlooked impact of glycolytic pathways and pyruvate metabolism on disease progression. The experiments are of high quality and properly controlled. Some points need addressing in order to improve the quality of the manuscript further, in particular some functional experiments are needed.

1. The authors have experimented with samples from patients in chronic phase. Presumably there are few LSC in these. Could the authors please confirm the percentage of LSC in these samples.
2. While there is a large amount of high throughput data, more functional and targeted assays should be performed to address the mechanism(s) in a more detailed manner. For instance: Do some of the discovered pathways impact quiescence? Does addition of glucose, glutamate or palmitate for example change quiescence/proliferation?
3. Please show that the human cells are still viable prior to performing these analyses.
4. Why would the authors primarily use K562 cells, which are blast phase CML cells, and do not truly represent real stem cells. A parallel between results with K562 and primary samples might suggest that the pathway isn't unique to stem cells but BCR-ABL1+ cells in general.
5. Figure S2J: Please verify any possible proliferation changes.
6. While imatinib is the standard of care for CML patients, why was no significance reached in S2A-B? Also, the text says $p = 0.081$ while the figure says a different value. Please comment.
7. Some experiments from figure 2, and S2 should be performed with another TKI, as IM can have effect on c-Kit too.
8. While the experiments shown in figure 4 are interesting, it is important to note that these experiments were conducted in the absence of stroma cells which could drastically change the observed effects. Could the authors repeat some of these experiments in the presence of stroma?
9. Please confirm some of the identified genes like CDK1 and TPX2 by Western blotting.
10. The concentration of MSDC-0160 is very high. Please comment.
11. Figures 2D-F: Seven days of treatment is a considerable amount of time and should kill most of the sensitive cells. Why did the authors treat for so long? Would this treatment be efficient at eradication resistant LSC also?
12. Figure S2K: There is no trend. A trend should have a p value of 0.1 or less.
13. AMPK induction should be shown convincingly with pAMPK
14. Possible upregulation of CD36 in the context of fatty acid oxidation should be discussed with respect to pyruvate anaplerosis.
15. If there is increased generation of glutamate via glutamine, does this affect the amount or activity of glutaminase (Figure 3)?

Reviewer #2:

Remarks to the Author:

What are the noteworthy results?

DEMONSTRATION OF THE ROLE OF PYRUVATE IN CML.

Will the work be of significance to the field and related fields?

YES

How does it compare to the established literature? If the work is not original, please provide relevant references.

VERY WELL, ON THE INNOVATIVE SIDE.

Does the work support the conclusions and claims, or is additional evidence needed?

YES, NO ADDITIONAL EVIDENCE NEEDED.

Are there any flaws in the data analysis, interpretation and conclusions?
NO

Do these prohibit publication or require revision?
NO

Is the methodology sound?
YES

Does the work meet the expected standards in your field?
YES

Is there enough detail provided in the methods for the work to be reproduced?
YES

Reviewer #3:

Remarks to the Author:

The authors propose that CML LSCs have significantly higher glucose metabolism compared to normal HSCs and that LSCs in CML would utilize pyruvate anaplerosis even in the presence of imatinib, but inhibition of pyruvate anaplerosis by the MPC1/2 inhibitor MSDC-0160 or by preventing mitochondrial import of pyruvate by CRISPR-Cas9-mediated KO of MPC1/2 does sensitize to imatinib treatment. In previous work (PMID: 28920959) the authors had already shown that oxidative phosphorylation was increased in CML LSCs, and LSCs were effectively eradicated by a combination of imatinib with the antibiotic tigecycline. Here, the authors start off by re-analyzing an earlier generated transcriptome datasets in which CML LSC transcriptomes were already analyzed. Besides earlier identified misregulated pathways such as MYC, p53 and E2F pathways, the authors now also identify alterations in glycolysis, oxidative phosphorylation, and fatty acid metabolism pathways, in line with studies by the authors themselves and others. An increase in steady state metabolite levels in CML CD34+ cells compared to CML CD34- cells for aspartate, malate and carnitine was already published earlier, but now the authors compare CML CD34+ against healthy CD34+ cells and again find an increase in steady state aspartate levels. 13C labeling studies using labeled glucose, palmitate and glutamine then reveal that it was particularly glucose metabolism that was enhanced in CML CD34+ cells compared to healthy CD34+ cells. Treatment with 2 uM imatinib for 48 hrs resulted in downregulation of 576 genes (linked to oxidative phosphorylation, glycolysis, and fatty acid oxidation pathways) and upregulation of 452 genes (linked to adipogenesis and p53 pathways) in 4 CML CD34+ samples. The authors conclude that PDH but not PC is downregulated upon imatinib treatment, and ultimately show that the remaining activity of this pathways maintains viability of CML CD34+ cells under imatinib treatment, but that co-inhibition with pyruvate import into the mitochondria would enhance imatinib efficacy. Overall, these are interesting studies aimed at a further elucidation of metabolic activities of CML CD34+ cells. It appears as if the main advance over their previous work in which CML CD34+ and CML LSC CD34+/CD38- cells were compared with CML CD34- cells (PMID: 28920959) is that now the authors compare CML CD34+ cells with healthy CD34+ cells. I find the notion that imatinib impacts on PDH but not so much on PC interesting, although there does appear to be quite some heterogeneity across samples and in some it appears as if PC activity is also inhibited. While it is clear that the lab has excellent expertise in CML LSC studies, it appears a bit unfortunate that most studies were conducted with CML CD34+ cells rather than focusing a bit deeper at the CML LSC CD34+/CD38- level. I have a number of specific comments/recommendations:

Fig 2: while in fig 1A-E previously generated data on CD34+/CD38- stem cell populations is analyzed, in fig 1F-J and fig 2 transcriptome and metabolome studies are performed in CML CD34+ cells, also +/- imatinib. The real problem in CML lies in CD34+/CD38- LSCs that are typically insensitive to imatinib. This population is only a minor fraction of the CML CD34+ population. How can we link these data to imatinib insensitive CML LSCs? Would it not be more informative to perform transcriptome and metabolome studies on those cells? For metabolome studies this would be more challenging with lower cell numbers available, but certainly feasible as

also previously shown by the authors. The authors state that "Collectively, these data show that TKI-mediated BCR-ABL inhibition causes downregulation of key metabolic pathways, although not sufficiently to prevent LSC survival" but in fact LSCs are not studied here. Also, the authors state that proliferation of these cells was inhibited at the dose of 2 μ M by determining the doubling time (Suppl fig 1A), $p = -.0659$), but that effects on cell viability were much more modest in sample-dependent ($p = 0.0809$). It appears as if both are certainly affected to similar degrees, with one sample (probably #5) responding less (please annotate the 4 individual samples in suppl. fig 4A and suppl 4B right panel). I could not find in the materials and methods section how population doubling was determined, but if this was simply done by counting viable cells I wonder how a viability loss of 10-30% is taken into account here and it looks like (part of) the loss in doubling time might in fact be explained by cell death. Based on this I do not think we can simply state that imatinib treatment did not impact on viability of CML CD34+ cells at all, I guess as expected, but that there is differential sensitivity in CVML CD34+ cells between patients, but most importantly the real imatinib insensitive CML LSCs are not captured by the transcriptome and metabolome studies presented here (even though the fig 2 legend title would suggest that).

In their previous Nature Med paper the authors did analyze metabolism in CD34+/CD38- HSCs and CML LSCs in ¹³C-palmitate tracing studies. In fact, there the conclusion was that incorporation of labeled palmitate in TCA intermediates was enhanced compared to normal CD34+/CD38- cells (Suppl fig 3) while now the authors conclude that there are no significant incorporations into TCA intermediates when using labeled palmitate (Fig. 1I). How can these differences be explained?

Are the four CML patients that are included here in fig 1g-j, fig, 2, fig 3 the same as were used in the Nat Med paper? Also in the Nat Med tracing studies with labeled glucose (CML CD34+ and CML CD34+/CD38-), and glutamine and palmitate (CML CD34+) were performed. In Supplementary table 1 I could not find which samples were used for fig 1g and 1i (not indicated) while I assume the same samples were used for the glucose, glutamine and palmitate labeling studies?

In Fig 2 and Supplemental Fig 2 the effects of imatinib show trends towards reduced mitochondrial activity and ATP production but many of these trends do not reach significance. Also, these studies are not performed on CD34+/CD38- cells. Yet, the authors conclude that "Overall, the results of the integrated omic analysis are consistent with TKI treatment causing a reduction of the main CML central carbon metabolism pathways."

By comparing the relative contributions of glucose, glutamine and palmitate to TCA intermediates (fig 3) the authors show that the majority of TCA intermediates is in fact generated from glutamine and palmitate, and both contribute equally except for citrate and glutamate to which palmitate contributes more. In comparison, only a small fraction of TCA intermediates is derived from glucose, although this might be partly explained by the relative high levels of palmitate that needed to be added. Moreover, imatinib did not impact on the relative contribution of palmitate, but did slightly reduce the contribution of glucose and glutamine. Upon imatinib treatment PDH expression was reduced, in line with the notion that PC activity was increased and that malate, citrate and aspartate were preferentially derived via the PC pathway in imatinib treated cells as shown by a reduction in m+2 incorporation while m+3 incorporation remained constant. Therefore, the authors conclude that "... these data suggest that PC-mediated pyruvate anaplerosis persists in imatinib treated CML CD34+ cells". I would agree with that, but as shown in Fig 2L basically all TCA enzymes are down upon imatinib treatment, not only PDH. Even though the TCA cycle is mainly driven by glucose under imatinib treatment, metabolism is overall reduced. One wonders what happens at the true CML LSC level, and also whether normal HSCs are not also able to use PC-mediated pyruvate anaplerosis to equal levels.

Upon MSDC-0160 treatment both the m+3 fraction (PC-dependent) as well as the m+2 fraction (PDH-dependent) was reduced, indicating that mitochondrial pyruvate important is important for both, as expected, upon imatinib treatment the reduction in m+2 was stronger compared to the effects on the m+3 Supplemental fig S3B. Still there appears to be quite a bit a variation there, with eg all samples showing a reduction in citrate m+3 as well (although does not reach significance) and also for 2 out of 4 cases there appears to be a strong reduction in malate m+3 and aspartate m+3, suggesting that the PC-dependent pathway is certainly affected by imatinib in those patients as well. "These results demonstrate that upregulated pyruvate anaplerosis in CML

LSCs, which is unaffected by imatinib (Fig. 2L) can be targeted by MSDC-0160." This appears to be somewhat of an overstatement. Moreover, it would be convincing to show this at the CML LSC level, and also that normal HSCs remain unaffected by MSDC-0160.

Supplemental fig 4: the m+3 fraction persists with MSDC-0160 treatment of K562 cells (B) but not really with UK-5099 (A) treatment, where we do see a reduction, how can this be explained? This also appears to be true in primary samples.

"Interestingly PC expression strongly associates with survival in paediatric AML but not adult AML (Fig. 4F, Supplementary Fig. S4E) while in adult AML, combined high expression of both MPC1 and 2 was associated with poorer survival (Fig. 4G, Supplementary Fig. S4E). Are the HRs in Suppl.fig 4E tested in multivariate analyses? Materials and methods does not provide any information. Univariate analyses these are not very informative. Also for the KM plots no information is provided as to which patients were included (eg were APL patient excluded from the TCGA cohort? Were only patient included with similar treatment protocols (7+3 and BM tx, or mixture of Tx versus no Tx?). These are all confounding factors that should be taken into account, without interpretation is difficult. And what would explain the difference between pediatric and adult AML? This is not eluded to at all.

Fig 5. Crispr-cas9-mediated KO of PC1 resulted in reductions in the m+5 labeled AKG and glutamate, but not m+2 and m+3 labelled TCA intermediates as was shown upon MPC1/2 inhibition. Why not? I understand that the m+5 fraction is derived from the condensation of PC-m+3 and PDH-m+2 but would we not also expect to see differences in m+2 and m+3 metabolite abundance? These do not appear to be affected at all, why not? And why is no data shown for malate, aspartate and citrate as in earlier figures?

Fig 5F-G/6C: the additional effects of the MSDC/imatinib combination do not appear to be very strong over MSDC alone (6C) or imatinib alone (5F-G).

We would like to thank all the reviewers for their constrictive comments as believe that by addressing them our manuscript has been substantially improved. Of significance, we have addressed comments on the relevance of our findings to the more primitive CML LSC (CD34⁺CD38⁻) population. In this regard, we now provide data demonstrating that the upregulated metabolism we see in CML CD34⁺ cells is clearly present in the rare CML CD34⁺CD38⁻ population. Due to the focus of our paper on glucose and pyruvate metabolism, and the limited availability of patient samples with large number of CD34⁺ cells (each CD34⁺38⁻ experiment requires minimum 20 million CD34⁺ cells at start), we have performed these challenging experiments by tracing ¹³C glucose.

We have highlighted all the changes in the manuscript accordingly and provide a point-by-point rebuttal to all Reviewer comments. These changes are highlighted in the manuscript and in the rebuttal letter as required.

REVIEWER COMMENTS Reviewer #1 - CML LSCs, imatinib resistance - (Remarks to the Author):

In the manuscript by Rattigan *et al.*, the authors have followed up on the role of metabolism and anaplerosis for the progression of BCR-ABL1-driven CML. The authors also show that an inhibitor of mitochondrial pyruvate carboxylase plus imatinib reduces CML LSC. The manuscript is well written and addresses a long-overlooked impact of glycolytic pathways and pyruvate metabolism on disease progression. The experiments are of high quality and properly controlled. Some points need addressing in order to improve the quality of the manuscript further, in particular some functional experiments are needed.

Response: We thank this reviewer for encouraging remarks regarding the research topic and the quality of our experiments.

1: The authors have experimented with samples from patients in chronic phase. Presumably there are few LSC in these. Could the authors please confirm the percentage of LSC in these samples.

Response: We appreciate that the CML CD34⁺ cell population is enriched for stem and progenitor cells. On average, and consistent with previous work, we find that 10-15% of CD34⁺ are CD38^{Low/-} and of the CD34⁺CD38⁻ population, approximately 50% are CD90⁺. As the only way to truly ascertain LSC frequency is by limiting dilution assay, we have addressed this point by repeating critical metabolomic assays in the CD34⁺CD38⁻ cell population (see rebuttal to Reviewer 3, point 1, for more detailed breakdown).

2: While there is a large amount of high throughput data, more functional and targeted assays should be performed to address the mechanism(s) in a more detailed manner. For instance: Do some of the discovered pathways impact quiescence? Does addition of glucose, glutamate or palmitate for example change quiescence/proliferation?

Response: We have already performed some of the experiments suggested and observed, using two separate patient samples, that CD34⁺ CML cells are very sensitive to glucose starvation,

resulting in high levels of cell death, compounding any additional functional assays (Rebuttal Fig. 1A). We also detected that physiological nutrient levels, or nutrient removal, will likely affect multiple pathways other than the TCA cycle. For instance, glucose is metabolised through the pentose phosphate pathway (Supplementary Fig. S1E [5 carbons to ATP] and glutamine is used to make glutathione; Rebuttal Fig. 1B) and is a proteinogenic amino acid. Thus, while it would be tempting to conclude based on viability that glucose is more important for viability, we cannot determine how much of this is due to its contribution to the TCA cycle.

Rebuttal Figure 1A-B

3. Please show that the human cells are still viable prior to performing these analyses.

Response: While viability was variable between samples (typically >60%) time points were chosen so that for LC-MS there was no significant difference in viability between experimental arms. For example, viability was >70% for all data included following the revision.

The following line has been added to supplementary materials (LC-MS sample preparation): *“Cells were counted, and viability calculated using a CASY counter with the following parameters: E-cur 7.5-22.5, N-cur 5.25-20.5. Timepoints were chosen to ensure equal viability within patient-derived samples for between experimental arms.”*

4. Why would the authors primarily use K562 cells, which are blast phase CML cells, and do not truly represent real stem cells. A parallel between results with K562 and primary samples might suggest that the pathway isn't unique to stem cells but BCR-ABL1+ cells in general.

Response: While we have identified the metabolic deregulation and confirmed vulnerability using robust PDX model, and overall aimed to do as many experiments as possible with primary samples, due to material/technical limitations we use cell lines for certain complementary studies. We now highlight K562 cell line as a model for PC-expressing CML line, and provide additional data in the

KCL22 cell line (new Supplementary Fig. S5C-F), which have low PC expression. Interestingly PC knockdown does not sensitise KCL22 cells to imatinib.

Relevant text has been added on page 16: *“Supporting this, knockout of PC in the KCL22 cell line, which have lower levels of PC, failed to increase sensitivity to imatinib (Supplementary Fig. S5C-F).”*

5. Figure S2J: Please verify any possible proliferation changes.

Response: This information is now in Supplementary Fig. S2A. Please note that there is patient dependent variance in both the doubling time and the effect of imatinib on doubling time (and viability; Supplementary Fig. S2B). We find that within samples the results are very consistent. For example, we had previously used CML sample CML#5 for a pilot experiment that was used to determine the timepoint and imatinib concentration used for Fig. 2-3 and Supplementary Fig. S2-3 and observed the same results.

6. While imatinib is the standard of care for CML patients, why was no significance reached in S2A-B? Also, the text says $p = 0.081$ while the figure says a different value. Please comment.

Response: We have corrected text to match Supplementary Fig. S2A-B. Our primary focus was to allow sufficient length of treatment while avoiding compromising integrity of the samples. While this was sample-dependent, we picked this timepoint as we didn't want to minimise levels in viability (i.e., this small non-significant change in viability is what we were aiming for).

7. Some experiments from figure 2, and S2 should be performed with another TKI, as IM can have effect on c-Kit too.

Response: We appreciate that imatinib affects other kinases such as c-Kit and have therefore performed additional experiment using the “cleaner” TKI, nilotinib (new data is provided in Supplementary Fig. S3F-G).

Relevant text has been added on page 12: *“Similar results were seen following nilotinib treatment, with reduction in total metabolite levels and ^{13}C -glucose incorporation into TCA cycle metabolites, suggesting that the effect is mediated through inhibition of BCR::ABL1, but not other kinases affected by imatinib such as c-Kit (Supplementary Fig. S3F-G)”*

8. While the experiments shown in figure 4 are interesting, it is important to note that these experiments were conducted in the absence of stroma cells which could drastically change the observed effects. Could the authors repeat some of these experiments in the presence of stroma?

Response: We appreciate that the presence of stromal cells might affect TCA cycle activity and/or the effect of MSCD-0160 on the inhibition of pyruvate oxidation. Therefore, we have performed additional glucose tracing experiments, using 3 separate patient samples, in the presence of stromal cells (new data presented in Supplementary Fig. S4E).

Relevant text has been added on page 14: *“While the dose required to fully inhibit labelling is relatively high (100 μ M), this is consistent with high FBS binding of the drug as previously reported⁴⁸. This inhibition was also achieved in the presence of stromal cells (Supplementary Fig. S4E).”*

9. Please confirm some of the identified genes like CDK1 and TPX2 by Western blotting.

Response: We have performed additional experiments using patient-derived cells as requested (Supplemental Fig. 2O). Note that these are not the same samples as for transcriptomic analysis and, as for most of the patient-sample data, there is sample dependent variation as expected (we had no material left for the samples used in the RNAseq experiment). We also used an earlier timepoint (24h vs 48h for RNAseq) to avoid deterioration in protein quality. Nevertheless, we have checked in 2 of the samples used for the LC-MS (48 hours) in this study to confirm that p-CRKL is reduced by imatinib (Supplemental Fig. 2O)

The following text has been added on page 9: *“At a protein level, albeit in different samples, the response in CDK1 levels varied in a sample dependent manner, while phospho-CRKL, which is immediately downstream of BCR::ABL1 signalling, was decreased by imatinib in the two patient samples tested (Supplementary Fig. 2O).”*

10. The concentration of MSDC-0160 is very high. Please comment.

Response: We agree that the dose required to fully inhibit MPC activity using MSDC-0160 is relatively high (50-100 μ M). A likely reason for the high dose used is that FBS can bind the drug. In line with this, the dose-effect we see is similar to Bader *et al* (Nat Met, pages 70-85, 2019) who showed that the presence of FBS explains discrepancy between the drug concentration required for oxidative respiration by Seahorse instrument (serum free conditions) and that used to inhibit cell growth *in vitro* (10% serum) (Bader *et al*, Supplementary Fig. 3G-H).

Additionally, the authors performed experiments using UK-5099 (Bader *et al*, Fig. 4G) and MSDC-0160 (Bader *et al*, Fig. 6D-E) which showed similar drug efficacy as MSDC-0160 treatment has in our study (Rebuttal Fig. 2)

Rebuttal Figure 2

11. Figures 2D-F: Seven days of treatment is a considerable amount of time and should kill most of the sensitive cells. Why did the authors treat for so long? Would this treatment be efficient at eradication resistant LSC also?

Response: We would like to clarify that we did not generate the 7-days dataset (E-MTAB-2594; Pellicano *et al*, Blood, 2018). We agree that 7-day treatment kills most of the sensitive cells and therefore enriches for primitive cells. However, we believe this is a relevant, high-quality dataset, and therefore interrogated it to complement our in-house 48 hours treatment dataset to establish if similar changes were observed.

12. Figure S2K: There is no trend. A trend should have a p value of 0.1 or less.

Response: We thank the review for this comment. This has now been corrected.

Relevant text has been added on page 10: *“While we detected a modest sample dependent increase in AMP/ATP ratio there was no significant change in ATP levels (Fig. 2G and Supplementary Fig. S2K).”*

13. AMPK induction should be shown convincingly with pAMPK

Response: We have performed Western Blot experiments using patient-derived cells (new data is presented in Supplementary Fig. 2O). The effect on p-AMPK is variable which may be expected for primary patient samples, with CML#20 showing an increase in p-AMPK following imatinib treatment. It is interesting that in CML#7 there is no visible change in p-AMPK, and also very modest change in AMP:ATP ratio (Fig 2G). Unfortunately, we did not obtain enough protein quantity to assess p-AMPK using sample CML#19. As mentioned above, for the revisions, there is no longer any material left for some of the samples used in the RNAseq and we used an earlier timepoint (24h vs 48h for RNAseq) to avoid deterioration in protein quality. Overall, we have toned down claims on this aspect of data.

Relevant text has been added on page 10: *“While we detected a modest sample dependent increase in AMP/ATP ratio there was no significant change in ATP levels (Fig. 2G and Supplementary Fig. S2K). Mitochondrial respiratory capacity (measured by FCCP-induced oxygen consumption rate) was reduced in imatinib-treated samples (Supplementary Fig. S2L). At a protein level, AMP-activated protein kinase (AMPK)⁴⁵ and phospho-AMPK levels were variable depending on patient sample (not detected in CML#9; Supplementary Fig. S2O).”*

14. Possible upregulation of CD36 in the context of fatty acid oxidation should be discussed with respect to pyruvate anaplerosis.

Response: In transcriptomics data, imatinib had no effect on CD36 transcript levels (P=0.3517), which is in line with unchanged fatty acid incorporation. We have highlighted this gene in volcano plot and added following line to text on page 9: *“Interestingly there was no change in CD36 (Supplementary Fig. S2D) which is highly upregulated in CML LSCs compared to normal HSCs (Supplementary Fig. S1A).”*

15. If there is increased generation of glutamate via glutamine, does this affect the amount or activity of glutaminase (Figure 3)?

Response: There was no imatinib-mediated change at a transcript level for glutaminase. While there were slight reductions in glutamine uptake and labelling, we didn't see changes at a transcript level for glutamine transporters or glutaminase, so our current data does not support this. We have added following line to text on page 9: *“Furthermore, no changes were detected in glutaminase or any of the glutamine transporters”*

Reviewer #2 - CML metabolism, LSCs - (Remarks to the Author):

GENERAL COMMENTS

The manuscript is written really well, the text flowing in a very straightforward way, with the relationships between consequential concepts kept extremely simple and clear. I wish to congratulate to the Authors for this. The quality and quantity of data reported in the manuscript are outstanding.

Response: We thank this reviewer for describing our manuscript as well written and clear, and for providing overall positive and encouraging comments regarding the quality of our manuscript.

Suggestions to improve the MS are listed hereafter.

1. Lines 80-82: "Several explanations for the TKI resistance of LSCs have been reported, including BCR-ABL kinase domain mutations, higher levels of BCR-ABL protein, BCRABL independent pathways". I would suggest to implement the text to include missing points and provide more complete information to the reader:

Response: We thank the reviewer for pointing this out. Additional relevant references and text has now been added on page 4: *"Firstly, mutations either within or outside BCR::ABL1 kinase domain²¹, amplification of the BCR::ABL1 gene²², quiescent cells with higher levels of BCR::ABL1 protein²³, have been reported. Also contributing to TKI-resistance, BCR::ABL1 independent pathways^{24, 25, 26}, suppression of BCR::ABL1 gene expression in hypoxia²⁷, and contribution of the bone marrow niche²⁶. Thirdly, enhanced activity of drug exporters has been reported to be critical to TKI-resistance in CML cell lines²⁸."*

(a) mutations of BCR/abl in tyrosine kinase domain;

Response: Addressed.

(b) amplification of BCR/abl gene (Haematologica 93:1718, 2008);

Response: Addressed.

(c) mutations outside BCR/abl gene determining BCR/Abl-independent survival and proliferation (secondary loss of "oncogene addiction");

Response: Addressed.

(d) primary BCR/Abl-independent Tessa's review Stem Cells 32:1373, 2014 (as opposed to the dependence of CML progenitors) (ref.s 22-24 of MS);

Response: Addressed.

(e) BCR/Abl protein suppression in BCR/abl gene-expressing cells (Leukemia 20:1291, 2006);

Response: Addressed.

(f) enhanced activity of drug exporters (PLoS-One 11(8):e0161470, 2016);

Response: Addressed.

(g) quiescence (ref. 21 of MS).

Response: Addressed.

As for (a) and (c), I would suggest to refer to Tessa's review Stem Cells 32:1373, 2014.

Response: Addressed.

2: Line 84 and lines 367-368 (and possibly elsewhere): the Authors may introduce mention to findings relative to the role of IL3 in glucose transport and metabolism (Differentiation 27:163, 1984. J. Biol. Chem. 272:17276, 1997;. Biochem. Pharmacol. 57:387, 1999; doi: 10.1016/s0006-2952(98)00267-6.) that, in connection to the IL3- mimetic effect of BCR/Abl (Blood 100:3731, 2002), is in my opinion relevant to their study. See also Oncogene 24:3257, 2005.

Response: This has been added in discussion as suggested, to put our findings in this context (page 18): *“Interestingly, IL-3 has been shown to promote glucose transport and metabolism while BCR::ABL1 (or RAS) can mimic this signalling and is IL-3 dependent”^{28, 54, 55, 56, 57, 58}*

3: Line 108: CD34+/CD38- are haematopoietic cell subsets which CONTAIN HSC or LSC; to state that they ARE HSC or LSC implies that ALL CELLS contained within that phenotypic fraction are stem cells, which is by far not the case. The statement should be toned down just to say that those cell fractions are ENRICHED WITH stem cells.

Response: We agree with this and have clarified this in the introduction (page 4, first paragraph) in in relevant text (top of page 6 as well as the sentence spanning page 7-8).

Line 351: The statement is correct, but it refers to all cell populations where proliferation is stimulated, independently of transition from the normal to cancer state, i.e. increased bioenergetics demand is, logically, not DIRECTLY related to neoplastic transformation. The Authors should take care of this concept throughout the manuscript. Pyruvate metabolism was shown to inhibit clonal expansion of normal clonogenic haematopoietic progenitors, but not their generation by pre-CFC, arguably stem cells (Exper. Hematol. 15:137, 1987. PMID: 3817047). Under this light too, it is very interesting the finding that “pyruvate carboxylation occurs in the mitochondria” and that this metabolic pathway, being unaffected by IM treatment, works normally in TKI-resistant CML cells. This makes me conclude, on one hand, that a stem cell subset (with specific metabolic adaptation features; pyruvate-resistant? or, more probably, pyruvate-dependent?) is actually selected under TKI pressure, and, on the other hand, that CML stem cells exhibit an OxPhos-oriented, mitochondria-driven metabolic asset (capable to process pyruvate). An Authors' comment on this would be appreciated.

Response: We've added the suggested reference with context (Pages 17-18): *“An important consideration is that rapid proliferating normal cells also undergo metabolic reprogramming. For*

*example, pyruvate metabolism has previously been shown to inhibit clonal expansion of normal clonogenic haematopoietic progenitor cells, which necessitates the inclusion of normal cells in metabolic studies*⁵³.

As for second point, imatinib reduces but does not completely shut down mitochondrial respiration. This residual activity may be supported by fatty acids, glutamine (albeit slightly reduced) and pyruvate. We could speculate that while total glucose-pyruvate flux is reduced, the pyruvate that is made can still enter TCA cycle by the MPC-PC axis. From previous experiments on CML, AML and CLL cell lines we see that the majority of glucose is converted to lactate (>90%) but that the fraction that enters the TCA cycle (via PDH or PC) contributes a large % of TCA cycle metabolite pools.

MINOR POINTS

There are flaws in the control of syntax, especially as far as the use of commas is concerned. The wrong use/lack of a comma often separates a verb from its subject, which is incorrect. A list of points to fix is as follows (but I haven't checked Materials and Methods and Legends to Figures accurately).

We thank the reviewer for the detailed review of our manuscript. We have provided Responses after each point:

1. Line 102: remove comma.

Response: Addressed.

2. Line 173: add comma after "that".

Response: Addressed.

3. Line 194: add a comma between "ratio" and "in".

Response: Addressed.

4. Line 231: add a comma between "that" and "for".

Response: Addressed.

5. Line 288: add a comma between "(Fig. 2L)" and "can".

Response: Addressed.

6. Line 290: remove comma after "AML" or add a comma between "while" and "in". The attribute of an adjective should be connected to the adjective itself with a dash; for example: patient-derived, isotope-assisted. Otherwise, patient becomes the subject of derived and isotope of assisted; indeed, this does not apply, for example, to "uniformly labelled", because an adverb cannot be the subject of a verb. The Authors sometimes put the dash, but they don't in the

majority of cases. A dash should be added in lines 29, 37, 233, 346, 358, 381, 384, 408 (I haven't checked Materials and Methods and Legends to Figures accurately).

Response: All these points have now been checked and corrected. Note that we have added new survival analysis which negates need for the correction on page 14.

7. Line 47: change "it" to "that resistance".

Response: Addressed.

8. Lines 355-356: I would suggest to change "proliferation of the leukaemic clone" to "leukaemic proliferation".

Response: Addressed.

9. Line 364: add a dash between "BCR-ABL" and "mediated" or change to "that BCR/Abl mediates".

Response: Addressed.

Reviewer #3 - Metabolomics, leukaemia - (Remarks to the Author):

The authors propose that CML LSCs have significantly higher glucose metabolism compared to normal HSCs and that LSCs in CML would utilize pyruvate anaplerosis even in the presence of imatinib, but inhibition of pyruvate anaplerosis by the MPC1/2 inhibitor MSDC-0160 or by preventing mitochondrial import of pyruvate by CRISPR-Cas9-mediated KO of MPC1/2 does sensitize to imatinib treatment. In previous work (PMID: 28920959) the authors had already shown that oxidative phosphorylation was increased in CML LSCs, and LSCs were effectively eradicated by a combination of imatinib with the antibiotic tigecycline. Here, the authors start off by re-analyzing an earlier generated transcriptome datasets in which CML LSC transcriptomes were already analyzed. Besides earlier identified misregulated pathways such as MYC, p53 and E2F pathways, the authors now also identify alterations in glycolysis, oxidative phosphorylation, and fatty acid metabolism pathways, in line with studies by the authors themselves and others. An increase in steady state metabolite levels in CML CD34+ cells compared to CML CD34- cells for aspartate, malate and carnitine was already published earlier, but now the authors compare CML CD34+ against healthy CD34+ cells and again find an increase in steady state aspartate levels. ¹³C labeling studies using labeled glucose, palmitate and glutamine then reveal that it was particularly glucose metabolism that was enhanced in CML CD34+ cells compared to healthy CD34+ cells. Treatment with 2 μ M imatinib for 48 hrs resulted in downregulation of 576 genes (linked to oxidative phosphorylation, glycolysis, and fatty acid oxidation pathways) and upregulation of 452 genes (linked to adipogenesis and p53 pathways) in 4 CML CD34+ samples. The authors conclude that PDH but not PC is downregulated upon imatinib treatment, and ultimately show that the remaining activity of this pathways maintains viability of CML CD34+ cells under imatinib treatment, but that co-inhibition with pyruvate import into the mitochondria would enhance imatinib efficacy. Overall, these are interesting studies aimed at a further elucidation of metabolic activities of CML CD34+ cells. It appears as if the main advance over their previous work in which CML CD34+ and CML LSC CD34+/CD38- cells were compared with CML CD34- cells (PMID: 28920959) is that now the authors compare CML CD34+ cells with healthy CD34+ cells. I find the notion that imatinib impacts on PDH but not so much on PC interesting, although there does appear to be quite some heterogeneity across samples and in some it appears as if PC activity is also inhibited. While it is clear that the lab has excellent expertise in CML LSC studies, it appears a bit unfortunate that most studies were conducted with CML CD34+ cells rather than focusing a bit deeper at the CML LSC CD34+/CD38- level. I have a number of specific comments/recommendations:

1. Fig 2: while in fig 1A-E previously generated data on CD34+/CD38- stem cell populations is analyzed, in fig 1F-J and fig 2 transcriptome and metabolome studies are performed in CML CD34+ cells, also +/- imatinib. The real problem in CML lies in CD34+/CD38- LSCs that are typically insensitive to imatinib. This population is only a minor fraction of the CML CD34+ population. How can we link these data to imatinib insensitive CML LSCs? Would it not be more informative to perform transcriptome and metabolome studies on those cells? For metabolome studies this

would be more challenging with lower cell numbers available, but certainly feasible as also previously shown by the authors.

Response: Firstly, we would like to thank the reviewer for the detailed and positive assessment of our manuscript. As the reviewer points out and we agree with, the CD34⁺CD38⁻ cell population is more enriched for the TKI-resistant CML LSCs. To address the reviewer comments, and increase the potential impact of our work, we have now performed additional metabolomic experiments using three separate CML and normal samples following sorting of the more primitive CD34⁺CD38⁻ cell population. As the reviewer points out, these experiments are technically challenging due to low number of cells available, so we were pleased to see the overall good quality of the data and consistency between each patient sample. Using ¹³C-glucose tracing, we reveal that while normal CD34⁺CD38⁻ cells are metabolically inactive when comes to TCA-cycle activity, CML CD34⁺CD38⁻ cells are significantly more active, and remarkably similar to the CD34⁺38⁺ cell population (new Figure 1K). This is indeed similar to what is seen at the transcript level and further underlines that primitive CML cells are metabolically distinct from normal cell.

Relevant text has been added on pages 7-8: *“While CD34⁺ enrich for progenitors and stem cells, CD34⁺CD38⁻ enriches further for primitive stem cells. Thus, we compared the fate of ¹³C-glucose in these two cell populations in normal and CML samples (Fig. 1K). While there was only modest amount of labelling in normal CD34⁺CD38⁻ and CD34⁺CD38⁺ cells, the labelling was significantly increased in both CML primitive populations. Notably there was only a slight, non-significant difference between CML CD34⁺CD38⁻ and CML CD34⁺CD38⁺ cells.”*

Moreover, we have also performed additional metabolomic experiments on CD34⁺CD38⁻ CML cells following single treatments with imatinib and MSDC-0160 (new Figures 3G and 4G).

Relevant text has been added on page 13: *“To investigate the relevance of these findings to the therapy resistant CD34⁺CD38⁻ population we quantified the effect of imatinib on ¹³C-glucose labelling in sorted cells (Fig 3G). While imatinib caused a statistically significant reduction in glucose incorporation in CML CD34⁺CD38⁻ cells, the relative PC/PDH activity increased in all three patient samples in this primitive population with a trend towards statistical significance (p-value = 0.0870).”*

In relation to MSDC-0160 treatments, relevant text has been added on pages 14-15: *“Moreover, we confirmed that inhibition of glucose oxidation was achieved in the primitive CD34⁺CD38⁻ population, to a similar extent as in the CD34⁺CD38⁺ progenitor population (Fig. 4G).”*

2. The authors state that “Collectively, these data show that TKI-mediated BCR-ABL inhibition causes downregulation of key metabolic pathways, although not sufficiently to prevent LSC survival” but in fact LSCs are not studied here.

Response: We hope that the addition of CD34⁺CD38⁻ data for Fig. 1K, 3G and 4G now supports this statement.

3. Also, the authors state that proliferation of these cells was inhibited at the dose of 2 μ M by determining the doubling time (Supl fig 1A), $p=-.0659$), but that effects on cell viability were much more modest en sample-dependent ($p=0.0809$). It appears as if both are certainly affected to similar degrees, with one sample (probably #5) responding less (please annotate the 4 individual samples in suppl. fig 4A and suppl 4B right panel).

Response: Samples are now annotated. Yes, CML#5 was less effected as also mentioned in our rebuttal to Reviewer 1, point 5.

4. I could not find in the materials and methods section how population doubling was determined, but if this was simply done by counting viable cells if wonder how a viability loss of 10-30% is taken into account here and it looks like (part of) the loss in doubling time might in fact be explained by cell death. Based on this I do not think we can simply state that imatinib treatment did not impact on viability of CML CD34+ cells at all, I guess as expected, but that there is differential sensitivity in CVML CD34+ cells between patients, but most importantly the real imatinib insensitive CML LSCs are not captured by the transcriptome and metabolome studies presented here (even though the fig 2 legend title would suggest that).

Response: this information has now been added to Methods (Cell culture): *“Cells were counted and viability calculated using a CASY counter with the following parameters to gate live cells: E-cur 7.5-22.5, N-cur 5.25-20.5. Doubling time over 48 hours was calculated by dividing the cell density into 4 times the seeding density ($4 \times 4 \times 10^5$ cells/mL = 16).”* For instance, sample grew from 4×10^4 cells/mL to 7.8×10^5 cells/mL in 48 hours and doubling time is calculated as 2.4 days. It is true that increased apoptosis could explain reduced proliferation. Therefore, we have modified the relevant part as follows (page 8): *“We applied the most commonly used TKI, imatinib, at a clinically relevant dose (2μ M)⁴², which inhibited cell doubling time (p -value = 0.0659; Supplementary Fig. S2A) in agreement with previous studies⁴³. It is possible that this increase is due to imatinib inducing a modest, sample-dependent increase in the level of apoptosis (p -value = 0.0809; Supplementary Fig. S2B).”* For the experiments in Figures 2-3 we picked as late as possible time point with minimal effects on cell survival, which was determined by pilot experiment using CML#5. Additionally, we hope that the new experiments that we have performed on CD34⁺CD38⁻ and CD34⁺CD38⁺ (Fig. 1K, 3G and 4G) helps extend our data from CD34⁺ cells to populations with greater enrichment for LSCs.

5. In their previous Nature Med paper the authors did analyze metabolism in CD34⁺/CD38⁻ HSCs and CML LSCs in ¹³C-palmitate tracing studies. In fact, there the conclusion was that incorporation of labeled palmitate in TCA intermediates was enhanced compared to normal CD34⁺/CD38⁻ cells (Supl fig 3) while now the authors conclude that there are no significant incorporations into TCA intermediates when using labelled palmitate (Fig. 1I). How can these differences be explained?

Response: For clarification, the ¹³C-palmitate tracing experiment in Kuntz *et al.* (Suppl. Fig 3B-D) is performed on CD34⁺ normal/CML cells (not CD34⁺CD38⁻). In this previous work, ¹³C-glucose tracing (not ¹³C-palmitate) was performed on a single patient CD34⁺CD38⁻ sample (Kuntz *et al.*, Suppl. Fig.

3G-I). However, the main difference can be explained by the type of analysis used. In the previous paper the focus was on steady state levels and fold changes, while we have examined fractional labelling that supports assessments of relative nutrient contribution. Given that the metabolite pool is higher in CML (Kuntz *et al.*), using fractional labelling is more relevant when measuring simultaneous contribution from all three nutrient inputs (glucose, glutamine, palmitate), which is the focus of our study.

As an example, and for the interest of the reviewer, the data in Kuntz *et al.* (Suppl. 3D) demonstrates that the steady state levels of unlabelled aspartate and total isotope labelling abundance is increased in CML CD34⁺ cells when compared with normal, which might be related to higher mitochondrial mass. However, the analysis applied in this study (Supplementary Fig. S1G) allows for the determination of the relative contribution of each of the three nutrients to the total activity of the pathway (Supplementary Fig. S3A). By analysing the relative contribution, we highlight that normal cells undergo similar rate of fatty acid oxidation, despite the smaller metabolite pool.

Relevant text has been added on page 7: *“However, given that the steady state levels of these metabolites are higher in CML cells than normal, increased fatty acid uptake via CD36 and subsequent oxidation may be needed to sustain these higher levels.”*

6. Are the four CML patients that are included here in fig 1g-j, fig 2, fig 3 the same as were used in the Nat Med paper? Also in the Nat Med tracing studies with labeled glucose (CML CD34⁺ and CML CD34⁺/CD38⁻), and glutamine and palmitate (CML CD34⁺) were performed. In Supplementary table 1 I could not find which samples were used for fig 1g and 1i (not indicated) while I assume the same samples were used for the glucose, glutamine and palmitate labeling studies?

Response: Overall, the same samples were not used for each experiment. We have now corrected and updated Supplementary Table 1. For clarification, we have used the same patient samples as used in Kuntz *et al.* for glucose and palmitate tracing experiments in this work. Specifically, CML#5, #7, #8 and #9 were used for most experiments. For Figure 1, both palmitate and glutamine used CML#1 and #2 while glucose and glutamine datasets both used CML#5, and also used CML#2, #4 and #6. Figures 2 and 3 have same patient samples, which is CML#5, #7, #8 and #9 while we were able to secure more of CML#5 and #7 for CD34⁺CD38⁻ work.

7. In Fig 2 and Supplemental Fig 2 the effects of imatinib show trends towards reduced mitochondrial activity and ATP production but many of these trends do not reach significance.

Response: We acknowledge this and have now toned down statements made regards to this (see also response to Reviewer 1 Point 13).

8. Also, these studies are not performed on CD34⁺/CD38⁻ cells. Yet, the authors conclude that “Overall, the results of the integrated omic analysis are consistent with TKI treatment causing a reduction of the main CML central carbon metabolism pathways.”

Response: We hope at the addition of metabolomics data for both CD34⁺CD38⁻ and CD34⁺CD38⁺ fractions for Figures 1K, 3G and 4G, which show that they are similarly metabolically active, addresses this point.

9. By comparing the relative contributions of glucose, glutamine and palmitate to TCA intermediates (fig 3) the authors show that the majority of TCA intermediates is in fact generated from glutamine and palmitate, and both contribute equally except for citrate and glutamate to which palmitate contributes more. In comparison, only a small fraction of TCA intermediates is derived from glucose, although this might be partly explained by the relative high levels of palmitate that needed to be added. Moreover, imatinib did not impact on the relative contribution of palmitate, but did slightly reduce the contribution of glucose and glutamine. Upon imatinib treatment PDH expression was reduced, in line with the notion that PC activity was increased and that malate, citrate and aspartate were preferentially derived via the PC pathway in imatinib treated cells as shown by a reduction in m+2 incorporation while m+3 incorporation remained constant. Therefore, the authors conclude that "... these data suggest that PC-mediated pyruvate anaplerosis persists in imatinib treated CML CD34⁺ cells". I would agree with that, but as shown in Fig 2L basically all TCA enzymes are down upon imatinib treatment, not only PDH. Even though the TCA cycle is mainly driven by glucose under imatinib treatment, metabolism is overall reduced. One wonders what happens at the true CML LSC level, and also whether normal HSCs are not also able to use PC-mediated pyruvate anaplerosis to equal levels.

Response: We have now performed additional experiments using more primitive CD34⁺CD38⁻ cells. These data are presented in new Fig. 3G and show good agreement with the data on CD34⁺ cells. Furthermore, we can now refer to Fig. 1K where it is apparent that there is essentially no PC activity in normal CD34⁺CD38⁻ cells (in contrast with imatinib-treated CML CD34⁺CD38⁻ cells; Fig. 3G). We have modified the statement mentioned above to highlight that the oxidative metabolism in treated cells is still decreased.

Relevant text has been added on page 13: *"Overall, these data suggest that PC-mediated pyruvate anaplerosis persists in imatinib-treated CML CD34⁺ cells and may contribute to residual TCA cycle activity."*

10. Upon MSDC-0160 treatment both the m+3 fraction (PC-dependent) as well as the m+2 fraction (PDH-dependent) was reduced, indicating that mitochondrial pyruvate important is important for both, as expected, upon imatinib treatment the reduction in m+2 was stronger compared to the effects on the m+3 Supplemental fig S3B. Still there appears to be quite a bit a variation there, with eg all samples showing a reduction in citrate m+3 as well (although does not reach significance) and also for 2 out of 4 cases there appears to be a strong reduction in malate m+3 and aspartate m+3, suggesting that the PC-dependent pathway is certainly affected by imatinib in those patients as well. "These results demonstrate that upregulated pyruvate anaplerosis in CML LSCs, which is unaffected by imatinib (Fig. 2L) can be targeted by MSDC-0160." This appears to be somewhat of an overstatement. Moreover, it would be convincing to show this at the CML LSC level, and also that normal HSCs remain unaffected by MSDC-0160.

Response: We have now rephrased the two lines below, highlighting the variation in m+3 as well as explaining the greater effect on PDH:

Relevant text has been added on page 12: *“Notably, PC relative to PDH contribution of glucose was increased in imatinib-treated cells, measured by the ratio of m+3 ($^{13}\text{C}_3$) to m+2 ($^{13}\text{C}_2$) labelling in malate, citrate, and aspartate (Fig. 3F). This is due to a more pronounced and consistent decrease in the PDH-derived fraction (m+2; Supplementary Fig. S3D) without a significant decrease in the PC-derived (m+3) fraction (only 2 out of 4 cases showed a consistent decrease in m+3), hence the increase in relative activity (PC/PDH) in all cases.”*

Relevant text has also been added on page 15: *“These results demonstrate that upregulated PC-mediated pyruvate anaplerosis in CML LSCs, which is less affected by imatinib than PDH-mediated pyruvate oxidation (Fig. 2L), can be targeted with MSDC-0160.”*

Additionally, we have now performed additional experiments using CML CD34⁺CD38⁻ cells following MSDC-0160 treatment. These data are presented in new Fig. 4G and again, show similar results as observed using CD34⁺ cells. Again, we can refer to Fig. 1K where it is apparent that there is essentially no contribution of glucose to the TCA cycle in normal CD34⁺CD38⁻ (therefore, without the evidence of any pyruvate contribution, it is not possible to assess the effect of MSDC-0160 on pyruvate anaplerosis in these cells).

11. Supplemental fig 4: the m+3 fraction persists with MSDC-0160 treatment of K562 cells (B) but not really with UK-5099 (A) treatment, where we do see a reduction, how can this be explained? This also appears to be true in primary samples.

Response: We thank the reviewer for pointing this out. We have realised that there was a mistake in the figure legends in the original manuscript which we have now corrected. The correct time-point for Supplementary Figure S4A (UK-5099/K562) is actually 48h (not 24h) which might explain the higher overall labelling (and m+3 fraction) in the untreated arm, when compared with Supplementary Figure S4B (MSDC-0160/K562) and Supplementary Figure S4C (MSDC-0160/CD34⁺ CML). Therefore, we think that part of the m+3 labelling in the intreated cells in Supplementary Figure S4A might come from the second cycle of the TCA-cycle (PDH-derived m+2 can become m+3 during the second cycle), and therefore by inhibited by UK-5099. As explained in point 12 below, we believe that the persistent m+3 fraction is likely contributed by cytoplasmic maleic enzyme, and therefore not affected by MPC inhibition.

12. “Interestingly PC expression strongly associates with survival in paediatric AML but not adult AML (Fig. 4F, Supplementary Fig. S4E) while in adult AML, combined high expression of both MPC1 and 2 was associated with poorer survival (Fig. 4G, Supplementary Fig. S4E). Are the HRs in Suppl.fig 4E tested in multivariate analyses? Materials and methods does not provide any information. Univariate analyses these are not very informative. Also for the KM plots no information is provided as to which patients were included (eg were APL patient excluded from the TCGA cohort? Were only patient included with similar treatment protocols (7+3 and BM tx, or mixture of Tx versus no Tx?). These are all confounding factors that should be taken into account,

without interpretation is difficult. And what would explain the difference between pediatric and adult AML? This is not eluded to at all.

Response: We thank the reviewer for these valuable and thorough suggestions. We have now replaced these results with multivariate analysis that considers treatment, transplant, FAB subtype and cytogenetics in updated Fig. 4F and Supplementary Fig. S4F. Here we have focused on adult AML where we feel our background knowledge supports this new analysis.

Relevant text has been added on page 15: *“Interestingly PC expression strongly associates with survival in adult AML with high-risk cytogenetics (Fig. 4F) with the difference just dropping outside significance threshold when excluding the M3 FAB subgroup (Supplementary Fig. S4F).”*

The relevant Methods section (Survival analysis) have been updated as well (pages 24-25).

13. Fig 5. Crispr-cas9-mediated KO of PC1 resulted in reductions in the m+5 labeled AKG and glutamate, but not m+2 and m+3 labelled TCA intermediates as was shown upon MPC1/2 inhibition. Why not? I understand that the m+5 fraction is derived from the condensation of PC-m+3 and PDH-m+2 but would we not also expect to see differences in m+2 and m+3 metabolite abundance? These do not appear to be affected at all, why not? And why is no data shown for malate, aspartate and citrate as in earlier figures?

Response: We have now added the requested plots in new Supplementary Fig. S5A-B. We would expect a reduction in m+3 but not m+2 (unless PDH is upregulated which may happen) as PC would be responsible for m+3 and PDH for m+2. We mostly see these changes, but the compounding factor is that for malate and aspartate there is a persisting fraction of m+3 which is likely derived from cytoplasmic maleic enzyme (which will not be reduced by PC ablation) and could be masking the decrease in PC derived labelling (Supplementary Fig. S5A-B).

Relevant text has been added on page 15: *“The m+2 fraction was unchanged or slightly increased while m+3 fractions were slightly decreased (Supplementary Fig. S5A-B), which is likely due to persistent cytoplasmic maleic enzyme activity seen in Supplementary Fig. S4A-B.”*

14. Fig 5F-G/6C: the additional effects of the MSDC/imatinib combination do not appear to be very strong over MSDC alone (6C) or imatinib alone (5F-G).

Response: We agree with this point. While we observed additive effect of imatinib and MSDC-0160 *in vitro*, this was not the case in the PDX experiments. We can speculate that the efficacy of MSDC-0160 on its own on the most primitive CML stem cells makes it difficult to detect any additional effect of imatinib at the drug concentrations used for *in vivo* work. Here we would also like to point out that we would not propose MSDC-0160 as a monotherapy, and the key message is that it is still effective even in the presence of TKI.

Reviewers' Comments:

Reviewer #1:

Remarks to the Author:

In this revised manuscript the authors have done a good job addressing this reviewer's comments. Small concerns are:

- 1) Line 175: $P=0.0659$ is a trend
- 2) Legend to figure 5: The authors mean nM, not nm, or microM, not microm.

Reviewer #2:

Remarks to the Author:

All the comments and suggestions from this reviewer appear to have met in the revised version of manuscript. However, one cannot avoid to note a couple of imprecisions at page 4, line 85 of revised MS:

1) the paper object of reference 27 doesn't shows the "suppression of BCR::ABL1 gene expression in hypoxia", but the fact that under very low oxygen tension BCR/Abl PROTEIN is suppressed, whereas the transcript (gene expression) is unchanged; this is exactly what enables low oxygen-resistant CML cells to maintain their full neoplastic nature while being completely refractory to TKi, due to the suppression of their molecular target; this point is in keeping with the findings of papers object of references 24 and 25;

2) the use of the term "hypoxia" should be avoided, as bone marrow environment (and stem cell niches in particular) are physiologically characterized by a relatively (very) low oxygen tension; this point is in keeping with the title of reference 26 ("Physiologic hypoxia..."); this reviewer suggest to shift to the use of the expression "(very) low oxygen tension".

Reviewer #3:

Remarks to the Author:

I would like to thank the authors for the additional data and modifications to the manuscript. I have no further comments.

REVIEWERS' COMMENTS

Reviewer #1 (Remarks to the Author):

In this revised manuscript the authors have done a good job addressing this reviewer's comments. **We are glad that revisions are to Reviewer 1's satisfaction.**

Small concerns are:

- 1) Line 175: $P=0.0659$ is a trend **Now corrected to mention trend**
- 2) Legend to figure 5: The authors mean nM, not nm, or microM, not microm. **Corrected.**

Reviewer #2 (Remarks to the Author):

All the comments and suggestions from this reviewer appear to have met in the revised version of manuscript. However, one cannot avoid to note a couple of imprecisions at page 4, line 85 of revised MS: **We are happy changes are satisfactory to Reviewer 2 and have addressed additional points below.**

1) the paper object of reference 27 doesn't shows the "suppression of BCR::ABL1 gene expression in hypoxia", but the fact that under very low oxygen tension BCR/Abl PROTEIN is suppressed, whereas the transcript (gene expression) is unchanged; this is exactly what enables low oxygen-resistant CML cells to maintain their full neoplastic nature while being completely refractory to TKi, due to the suppression of their molecular target; this point is in keeping with the findings of papers object of references 24 and 25; **This has been corrected 'suppression of BCR::ABL1 protein level but not gene expression'.**

2) the use of the term "hypoxia" should be avoided, as bone marrow environment (and stem cell niches in particular) are physiologically characterized by a relatively (very) low oxygen tension; this point is in keeping with the title of reference 26 ("Physiologic hypoxia..."); this reviewer suggest to shift to the use of the expression "(very) low oxygen tension". **This has been replaced by 'low oxygen tension typical of the bone marrow niche'**

Reviewer #3 (Remarks to the Author):

I would like to thank the authors for the additional data and modifications to the manuscript. I have no further comments. **We are glad that new data are to Reviewer 3's satisfaction.**